# Unsupervised Learning for Combinatorial Optimization Needs Meta Learning

**Haoyu Wang**[1]**, Pan Li**[1,2]
1. Department of Electrical and Computer Engineering, Georgia Institute of Technology
2. Department of Computer Science, Purdue University
`hwang3028@gatech.edu, panli@gatech.edu, panli@purdue.edu`

## Abstract

A general framework of unsupervised learning for combinatorial optimization (CO) is to train a neural network whose output gives a problem solution by directly optimizing the CO objective. Albeit with some advantages over traditional solvers, current frameworks optimize an averaged performance over the distribution of historical problem instances, which misaligns with the actual goal of CO that looks for a good solution to every future encountered instance. With this observation, we propose a new objective of unsupervised learning for CO where the goal of learning is to search for good initialization for future problem instances rather than give direct solutions. We propose a meta-learning-based training pipeline for this new objective. Our method achieves good performance. We observe that even the initial solution given by our model before fine-tuning can significantly outperform the baselines under various evaluation settings including evaluation across multiple datasets, and the case with big shifts in the problem scale. The reason we conjecture is that meta-learning-based training lets the model be loosely tied to each local optimum for a training instance while being more adaptive to the changes of optimization landscapes across instances. [1]

## 1 Introduction

Combinatorial optimization (CO), aiming to find out the optimal solution from discrete search space, has a pivotal position in scientific and engineering fields (Papadimitriou & Steiglitz, 1998; Crama, 1997). Most CO problems are NP-complete or NP-hard. Conventional heuristics or approximation requires insightful comprehension of the particular problem. Starting from the seminal work from Hopfield & Tank (1985), researchers apply neural networks (NNs) (Smith, 1999; Vinyals et al., 2015) to solve CO problems. The motivation is that NNs may learn heuristics through solving historical problems, which could be useful to solve similar problems in the future.

Many NN-based methods (Selsam et al., 2018; Joshi et al., 2019; Hudson et al., 2021; Gasse et al., 2019; Khalil et al., 2016) require optimal solutions to the CO problem as supervision in training. However, optimal solutions are hard to get in practice and the obtained model often does not generalize well (Yehuda et al., 2020). Methods based on reinforcement learning (RL) (Mazyavkina et al., 2021; Bello et al., 2016; Khalil et al., 2017; Yolcu & Póczos, 2019; Chen & Tian, 2019; Yao et al., 2019; Kwon et al., 2020; 2021; Delarue et al., 2020; Nandwani et al., 2021) do not need labels while they often suffer from notoriously unstable training. Recently, unsupervised learning methods have attracted much attention (Toenshoff et al., 2021; Amizadeh et al., 2018; Yao et al., 2019; Karalias & Loukas, 2020; Wang et al., 2022). A common strategy of these methods is to design an NN whose output gives a solution to the CO problem and then train the NN via gradient descent by directly optimizing the CO objectives over a set of training instances. This strategy is superior in its faster training, good generalization, and strong capability of dealing with large-scale problems.

Despite the prominent progress, current unsupervised learning methods always optimize NNs towards an *averaged good* performance over training instances. This means even if a testing instance comes from the same distribution of the training instances, the solution to this single instance may not have good quality, let alone the case when the testing instance is out-of-distribution (OOD). This

---

[1]Our code is available at: `https://github.com/Graph-COM/Meta_CO`

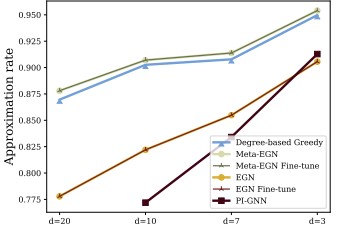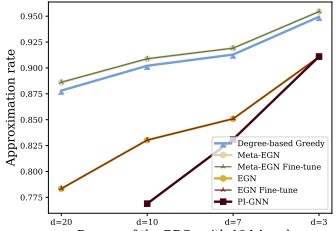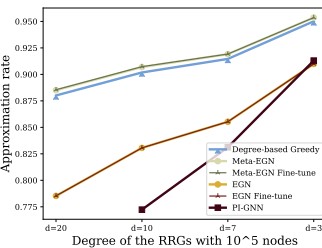

Figure 1: Approximation Rates of different methods in the MIS problem. Meta-EGN and EGN (Karalias & Loukas, 2020) are trained on RRGs with 1000 nodes and with node degree randomly sampled from 3, 7, 10, 20. Meta-EGN and EGN are evaluated over larger RRGs with $10^3 \sim 10^5$ nodes. More details about the setting are in Secs. 5.1 and 5.4. Meta-EGN outperforms DGA (Angelini & Ricci-Tersenghi, 2019) by about $0.3\% - 0.5\%$ in approximation rates on average.

induces a concern when we apply NNs in practice because practical problems often expect to have a good solution to every encountered instance. For example, allocating surveillance cameras is crucial for each-time exhibition in every art gallery. Solvers when applied to this problem (O'rourke, 1987; Yabuta & Kitazawa, 2008) should output a good solution every time. Traditional CO solvers are designed toward this goal. However, it is time-consuming and unable to learn heuristics from historical instances. So, can we leverage the benefit of learning from history with the goal of achieving an instance-wise good solution instead of an averaged good solution?

This motivates us to study a new formulation of unsupervised learning for CO. We regard the objective of learning from history as to search for a good initialization for each future instance rather than give a direct solution. Since in practice, future instances are unavailable during the training stage, we propose to view each training instance as a pseudo-new instance for the rest training instances. Then, our learning objective is to learn a good initialization of this model, such that further optimization of the initialization could achieve good solutions on each of these pseudo-new instances. We observe meta learning is suitable to implement this idea and propose to adopt MAML (Finn et al., 2017) in our training pipeline as a proof of concept. Note that the step of optimization on each pseudo-new instance shares a similar spirit with fine-tuning a model over each down-streaming task as traditional meta learning does. However, each task in our case corresponds to optimization over each training instance.

We name our method Meta-EGN by extending the previous framework EGN (Karalias & Loukas, 2020) via meta learning. Our key observation is that with this new objective, even the initial solution given by Meta-EGN (before fine-tuning on a test instance) is substantially better than the solution given by EGN and other methods that optimize the averaged performance over training instances. Our conjectured reason is that the new objective, by taking into account fine-tuning the model over new instances, trains the model to avoid being trapped into a local minimum induced by each training instance while being more adaptive to the changes of optimization landscapes across instances.

We demonstrate the benefits of Meta-EGN via experiments within three benchmark CO problems (max clique, vertex cover, and max independent set) on multiple synthetic graphs and three real-world graph datasets, with the number of nodes ranging from 100 to 5000. Meta-EGN significantly outperforms state-of-the-art learning-based baselines (Karalias & Loukas, 2020; Toenshoff et al., 2021), greedy algorithms, and the commercial CO solver Gurobi9.5 (Gurobi Optimization, 2022) in most cases. Meta-EGN also shows super OOD generalization performance when the training and test datasets are different or have graphs of entirely different sizes.

Moreover, recently, Angelini & Ricci-Tersenghi (2022) have shown that the learning-based method in (Schuetz et al., 2022) could not achieve comparable results with the degree-based greedy algorithm (DGA) (Angelini & Ricci-Tersenghi, 2019) in the max independent set (MIS) problem on large-scaled random-regular graphs (RRGs), which raises attentions from machine learning community. We observe the issues come from two aspects: (1) graph neural networks (GNNs) used to encode the regular graph suffer from the node ambiguity issue due to their limited expressive power (Xu et al., 2019); (2) the model in (Schuetz et al., 2022) did not learn from history but was directly optimized over each testing case, which tends to be trapped into a local optimum. By addressing these two issues, Meta-EGN can consistently outperform DGA while maintaining the same time complexity to generate solutions. Fig. 1 show the results.

## 2 RELATED WORK

In the following, we review two groups of works: unsupervised learning for CO and meta learning.

Previous works on unsupervised learning for CO have studied max-cut (Yao et al., 2019) and TSP problems (Hudson et al., 2021), while these works depend on carefully selected problem-specific objectives. Some works have investigated satisfaction problems (Amizadeh et al., 2018; Toenshoff et al., 2019). Applying these approaches to general CO problems requires problem reductions. The works most relevant to us are (Karalias & Loukas, 2020), (Wang et al., 2022) and (Schuetz et al., 2022). Karalias & Loukas (2020) propose an unsupervised learning framework EGN for general CO problems based on the Erdős' probabilistic method, which bonds the quality of the final solutions with probability. Wang et al. (2022) generalize EGN and prove that if the CO objective can be relaxed into an entry-wise concave form, a solution of good quality can be deterministically achieved. This further inspires the design of proxy objectives for CO problems that may not have closed-form objectives, such as those in circuit design. Schuetz et al. (2022) have recently extended EGN to large-scale max independent set problems on random-regular graphs.

Meta learning is proposed to learn hyper-parameters or initialization from historical tasks and achieve fast adaption to new tasks. Finn et al. (2017) propose model-agnostic meta learning (MAML), which aims to obtain good parameter initialization and be accommodated to few-shot learning with limited steps of fine-tuning. Nichol et al. (2018) accelerate MAML by adopting first-order approximation on the gradient estimation. Rajeswaran et al. (2019) introduce implicit-MAML that adopts an objective with fine-tuning till the stationary points on new tasks. Implicit-MAML does not fit our case because we try to avoid long-time fine-tuning. Hsu et al. (2018) study unsupervised learning under the meta learning framework and focus exclusively on vision tasks. To the best of our knowledge, our work is the first to apply meta learning to unsupervised learning for CO.

## 3 PRELIMINARIES: NOTATIONS AND PROBLEM FORMULATION

**Combinatorial Optimization on Graphs.** We follow the settings considered in (Karalias & Loukas, 2020; Wang et al., 2022; Schuetz et al., 2022) and study CO problems on graphs whose solutions can be represented as a subset of nodes of the input graph instance, although our method could be applied to a broader range of problems. Suppose $\mathcal{G}$ is the universe of graph instances. Let $G(V, E) \in \mathcal{G}$ denote a graph instance where $V = \{1, 2, ..., n\}$ is the node set and $E$ is the edge set. Let $X = (X_i)_{1 \leq i \leq n} \in \{0, 1\}^n$ denote the discrete optimization variables defined on $V$, where $X_i = 1$ denotes that node $i$ is selected in the output node subset. A CO problem on $G$ consists of a cost function $f(\cdot; G) : \{0, 1\}^n \to \mathbb{R}_{\geq 0}$ and a feasible set $\Omega \subseteq \{0, 1\}^n$ that stands for the finite set of all feasible $X$'s, and asks to solve

$$\min_{X \in \{0,1\}^n} f(X; G) \qquad \text{s.t.} \quad X \in \Omega \tag{1}$$

**Unsupervised Learning for CO.** The Erdös-Goes-Neural (EGN) framework of unsupervised learning for CO proposed in (Karalias & Loukas, 2020) is reviewed as follows. Here, we use the notation system in a follow-up work (Wang et al., 2022) as it is more clear. Learning for CO problem is to learn an algorithm $\mathcal{A}_\theta(\cdot) : \mathcal{G} \to \{0, 1\}^n$ typically parameterized by an NN where $\theta$ denotes the parameters of the NN such that given a graph instance $G$, $X = \mathcal{A}_\theta(G)$ gives a solution of Eq. 1. In practice, directly optimizing the parameters $\theta$ is hard in general.

Therefore, we may consider a relaxed cost function $f_r(\cdot; G) : [0, 1]^n \to \mathbb{R}_{\geq 0}$ where $f_r(X; G) = f(X; G)$ on any discrete points $X \in \{0, 1\}^n$ and a relaxed constraint $g_r(\cdot; G) : [0, 1]^n \to \mathbb{R}_{\geq 0}$ where $\{X \in \{0, 1\}^n : g_r(X; G) = 0\}$ and $\{X \in \{0, 1\}^n : g_r(X; G) \geq 1\}$ defines the feasible set $\Omega$ and the infeasible set $\Omega^c$ respectively. Also, suppose the NN in $\mathcal{A}_\theta$ can give soft solutions $\bar{X} \in [0, 1]^n$. Then, we may train $\theta$ by minimizing a label-independent loss function:

$$\min_\theta l(\theta; G) \triangleq f_r(\bar{X}; G) + \beta g_r(\bar{X}; G), \quad \bar{X} = \mathcal{A}_\theta(G), \text{ for some } \beta > 0. \tag{2}$$

The significant observation made by (Wang et al., 2022), which generalizes the argument in (Karalias & Loukas, 2020), is a type of performance guarantee on the condition that $f_r$ and $g_r$ are entry-wise concave, which is satisfied in all the cases studied in this work: If the loss that achieves $l(\theta; G) < \beta$ for some $\beta > \max_{X \in \{0,1\}^n} f(X; G)$, then the discrete solution $X$ obtained by rounding the soft solution $\bar{X} = \mathcal{A}_\theta(G)$ according to Def. 1 is feasible $X \in \Omega$ and of good quality $f(X; G) \leq l(\theta; G)$.

---

**Algorithm 1** Train Meta-EGN and Test Meta-EGN with/without Fine-tuning

---

**Require:** Training instances $\Xi = \{G_1, G_2, ..., G_m\}$; Hyperparameters: $\alpha, \gamma$.
1: Randomly initialize $\theta^{(0)}$
2: **for** each randomly sampled mini-batch $B_j \subset \Xi$, $j = 0, 1, ..., K-1$ **do**   $\triangleright$ Training starts
3:  For each $G_i \in B_j$, compute the adapted parameter: $\theta_i^{(j)} = \theta^{(j)} - \alpha \nabla_{\theta^{(j)}} l(\theta^{(j)}; G_i)$
4:  Update: $\theta^{(j+1)} \leftarrow \theta^{(j)} - \gamma \nabla_{\theta^{(j)}} \sum_{G_i \in B_j} l(\theta_i^{(j)}; G_i)$
5: **end for**
6: **return** $\theta \leftarrow \theta^{(K)}$               $\triangleright$ Training ends
7: For a given testing instance $G'$:           $\triangleright$ Testing starts
8: **if** fine-tuning is allowed **then**
9:  Fine-tune the parameters: $\theta_{G'} \leftarrow \theta - \alpha \nabla_\theta l(\theta; G')$
10:  Use Def. 1 to round the relaxed solution given by $\mathcal{A}_{\theta_{G'}}(G')$   $\triangleright$ With fine-tuning
11: **else**
12:  Use Def. 1 to round the relaxed solution given by $\mathcal{A}_\theta(G')$   $\triangleright$ Without fine-tuning
13: **end if**                 $\triangleright$ Testing ends

---

**Definition 1** (Rounding). *For a soft solution $\bar{X} \in [0,1]^n$ and an arbitrary order of the entries (w.l.o.g 1,2,...,n), fix all the other entries unchanged and round $\bar{X}_i$ into 0 or 1 as $X_i = \arg\min_{j=0,1} f_r(X_1, ..., X_{i-1}, j, \bar{X}_{i+1}, ..., \bar{X}_n) + \beta g_r(X_1, ..., X_{i-1}, j, \bar{X}_{i+1}, ..., \bar{X}_n)$, replace $\bar{X}_i$ with $X_i$ and repeat this operation until all the entries are discrete.*

## 4 META LEARNING FOR ERDÖS GOES NEURAL (META-EGN)

The above performance guarantee lays the theoretical foundation for EGN. However, the following practical issue motivates us to incorporate meta learning into EGN.

### 4.1 MOTIVATION: WHAT NEEDED IS LEARNING FOR INSTANCE-WISE GOOD SOLUTIONS

It is often time-consuming to perform online optimization of $l(\theta; G)$ for each encountered instance $G$. This also mismatches the goal of learning, i.e., learning heuristics from history/data. Therefore, a pipeline commonly adopted is as follows. Suppose there is a set of training instances $G_i$, $1 \leq i \leq m$, IID sampled from a distribution $\mathbb{P}_\mathcal{G}$. We optimize $\theta$ by following

$$\min_\theta \sum_{i=1}^m l(\theta; G_i), \tag{3}$$

which is similar to empirical risk minimization (ERM) in standard supervised learning. When a test instance $G$ appears, we apply the learned $\mathcal{A}_\theta$ to get a soft solution and round it to the final solution.

This pipeline cannot guarantee the quality for this instance $G$. Even if the training instances $G_i$, $1 \leq i \leq m$ are in a large quantity (so in-distribution generalization is not a problem), and even if the test instance $G$ also follows $\mathbb{P}_\mathcal{G}$, we may not guarantee a low $l(\theta; G)$ for one particular $G$ because ERM only guarantees a low averaged performance $\mathbb{E}_{G \sim \mathbb{P}_\mathcal{G}}[l(\theta; G)]$. This issue may also violate the condition to have a performance guarantee as reviewed in Sec. 3, as it is instance-wise. Here, we highlight that in practice even the minimal averaged loss $\min_\theta \mathbb{E}_{G \sim \mathbb{P}_\mathcal{G}}[l(\theta; G)]$ is often strictly greater than averaged instance-wise minimal loss $\mathbb{E}_{G \sim \mathbb{P}_\mathcal{G}}[\min_\theta l(\theta; G)]$, because practical NNs are not expressive enough to remember the optimal solution to every instance.

Unfortunately, many practical CO problems actually expect *instance-wise good solutions*. This is because every instance in practice is crucial. A terrible solution for one instance may raise a security issue (e.g., the surveillance-camera allocation problem) or cause huge economic losses (e.g., the routing problem in a transportation system). With this observation, our work is to address the problem by studying *unsupervised learning for instance-wise good solutions to CO problems*.

### 4.2 TRAINING TOWARDS INSTANCE-WISE OPTIMALITY VIA META LEARNING

Our idea to address the problem is to regard the goal of learning from history as to search for good initialization for future instances rather than give direct solutions. Such good initialization

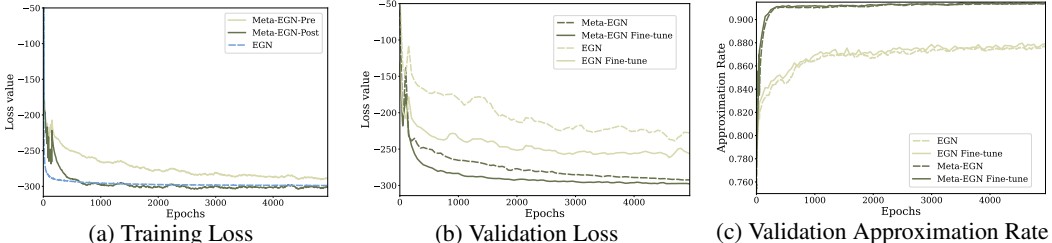

(a) Training Loss          (b) Validation Loss          (c) Validation Approximation Rate

Figure 2: Training/validating dynamics of Meta-EGN and EGN Karalias & Loukas (2020) for the MIS problem. Detailed experiment settings follow Sec. 5.1.

can be quickly fine-tuned by further optimizing the model for each instance, which ultimately gives instance-wise good solutions. However, in practice, we do not have access to any future/test instances. So, can we just use historical/training instances to implement the above idea? Our strategy is to view each training instance $G_i$ as a pseudo-test instance to test and optimize the quality of initialization given by the model. Specifically, this strategy gives us an objective

$$\min_{\theta} \sum_{i=1}^{m} \tilde{l}_i(\theta), \quad \text{where } \tilde{l}_i(\theta) = \min_{\theta_i} l(\theta_i; G_i) \text{ with } \theta_i = \theta \text{ as initialization.} \quad (4)$$

Eq. 4 has some abuse of notations. The minimum $\tilde{l}_i(\theta)$ depending on the initialization $\theta$ is because of the non-convex nature of $\min_{\theta_i} l(\theta_i; G_i)$, where the initialization $\theta_i = \theta$ matters significantly.

We further simplify Eq. 4 with some practical consideration. In fact, we may not allow further optimizing $\theta$ with so many gradient-descent steps for each instance, especially during the online test stage. As a proof of concept, we consider the case with only one-step gradient descent, which already gives good empirical results. Specifically, our training objective follows

$$\textbf{Our Objective: } \min_{\theta} \sum_{i=1}^{m} l_i(\theta), \quad \text{where } l_i(\theta) = l(\theta_i; G_i) \text{ with } \theta_i = \theta - \alpha \nabla_{\theta} l(\theta; G_i). \quad (5)$$

Here, $\theta$ is to give a good initialization $\mathcal{A}_{\theta}(G_i)$ over each instance $G_i$ while $\theta_i$ is with one-step fine-tune to achieve a $G_i$-specified good solution $\mathcal{A}_{\theta_i}(G_i)$.

Optimization in Eq. 5 can be implemented via the meta learning pipeline MAML (Finn et al., 2017). We name the obtained model Meta-EGN and summarize its training and testing in Alg. 1. In step 3, Meta-EGN performs the one-step gradient descent on each training instance. Note that we consider two testing cases with or without fine-tuning because the latter saves much inference time. A simple extension of Theorem 1 in (Wang et al., 2022) gives a performance guarantee for Meta-EGN in Theorem 1 as follows. Here, for a test instance $G$, we even allow $l(\theta; G)$ to violate the original condition $l(\theta; G) < \beta$ in (Wang et al., 2022) to some extent. After one-step fine-tuning in step 9, the performance guarantee is still achievable. The detailed proof is in Appendix. A.

**Theorem 1** (Performance Guarantee). *Suppose the relaxations $f_r$ and $g_r$ are entry-wise concave as required in (Wang et al., 2022). Let $\theta$ denote the learned parameter after training. Given a test instance $G$, suppose locally $l(\cdot; G)$ is $L$-smooth at $\theta$, i.e., $\|\nabla_{\theta'} l(\theta'; G) - \nabla_{\theta} l(\theta; G)\| \leq L\|\theta' - \theta\|$ for all $\theta'$ that satisfies $\|\theta' - \theta\| \leq \epsilon$. Then, if $\underline{l(\theta; G) < \beta + \triangle}$ (even if $l(\theta; G) \geq \beta$), for any $\alpha \in (0, 2/L)$ Meta-GNN with one-step finetuning $\underline{\text{outputs a feasible solution } X \text{ of good quality}}$ $f(X; G) \leq l(\theta; G) - \triangle$. Here, $\triangle = \|\nabla_{\theta} l(\theta; G)\|\epsilon + \frac{1}{2L\alpha^2 - 4\alpha}\epsilon^2$ if $\epsilon < \alpha\|\nabla_{\theta} l(\theta; G)\|$ or $\triangle = (\alpha - \frac{L\alpha^2}{2})\|\nabla_{\theta} l(\theta; G)\|^2$ o.w..*

To better understand Meta-EGN, we show its training/testing dynamics in Fig. 4.2. As we expected, the training loss of EGN is somewhere in-between the losses of Meta-EGN before and after the fine-tuning step. Training EGN is stabler and converges faster than training Meta-EGN. However, what is unexpected is that in validation, Meta-EGN has a much lower loss and achieves much better performance than EGN even before fine-tuning.

This implies that Meta-EGN holds better generalization than EGN. We conjecture the reasons are as follows. First, the optimization landscape for CO problems is extremely non-convex (Mezard & Montanari, 2009) due to the intersected feasible-infeasible regions and the high penalty coefficient

Table 1: Comparison between different unsupervised frameworks. $G$ denotes the test instance and $G_i$, $1 \leq i \leq m$ are training instances. The standard EGN pipeline does not adopt any fine-tuning.

|  | EGN (Karalias & Loukas, 2020) | P-I GNN (Schuetz et al., 2022) | Meta-EGN (Ours) | Classical Solver Gurobi Optimization (2022) |
|---|---|---|---|---|
| Obj. to optimize the NN | $\sum_{i=1}^{m} l(\theta; G_i)$ | $l(\theta; G)$ | $\sum_{i=1}^{m} l(\theta - \nabla_\theta l(\theta; G_i); G_i)$ | $f(X; G)$ s.t. $X \in \Omega$ |
| Training or not | Yes | No | Yes | No |
| Fine-tune timing | No | Long | Short/No | Long |
| Generalization | Good | - | Better | - |

Table 2: The discrete objectives (Eq. 1) and their relaxations (Eq. 2) for the three problems to be studied.

| MC | Discrete Obj. | $\max_X \sum_{1 \leq i \leq n} X_i$     s.t.    $(i,j) \in E$ if $X_i, X_j = 1$ |
|---|---|---|
|  | Relaxation | $l_{\text{MC}}(\theta; G) \triangleq -(\beta + 1) \sum_{(i,j) \in E} \bar{X}_i \bar{X}_j + \frac{\beta}{2} \sum_{i \neq j} \bar{X}_i \bar{X}_j$ |
| MVC | Discrete Obj. | $\min_X \sum_{1 \leq i \leq n} X_i$     s.t.    $X_i + X_j \geq 1$ if $(i,j) \in E$ |
|  | Relaxation | $l_{\text{MVC}}(\theta; G) \triangleq \sum_{1 \leq i \leq n} \bar{X}_i + \beta \sum_{(i,j) \in E} (1 - \bar{X}_i)(1 - \bar{X}_j)$ |
| MIS | Discrete Obj. | $\max_X \sum_{1 \leq i \leq n} X_i$     s.t.    $X_i X_j = 0$ if $(i,j) \in E$ |
|  | Relaxation | $l_{\text{MIS}}(\theta; G) \triangleq -\sum_{1 \leq i \leq n} \bar{X}_i + \beta \sum_{(i,j) \in E} \bar{X}_i \bar{X}_j$ |

Figure 3: Performance v.s. hyper-parameter $\rho$ of the RB model

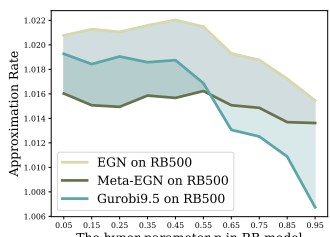

$\beta$. EGN that has low losses for training instances may give a high loss even when the optimization landscape is just slightly shifted (from training to a test instance). However, the parameters of Meta-EGN are loosely tied to a local minimum for each training instance. Instead, those parameters, being aware of follow-up instance-wise fine-tuning steps, are likely to fall into some location close to a local minimum for each instance while being not trapped in any one of them, which makes the model robust to landscape shifts across instances. Second, a CO problem could vary a lot across graph instances even for those generated from the same distribution, especially when the instances are large. So, it is reasonable to view the problem over each instance as a separate but relevant task. Meta learning has shown good generalization when data distributions shift across tasks, which has empirical evidence in CV and NLP applications (Jeong & Kim, 2020; Conklin et al., 2021).

As a summary, we provide a comparison between different unsupervised frameworks to solve CO problems in Table 1. Note that PI-GNN (Schuetz et al., 2022) is directly fine-tuned on each test instance without training so the fine-tuning time is long. Also, although PI-GNN also pursues instance-wise good solutions, its performance could be bad because it does not learn from training instances. The instance-wise solutions could be just bad local minima.

## 5 EXPERIMENTS

We study three CO problems, namely *max clique (MC)* to find the largest set of nodes where each pair of nodes are connected, *minimum vertex covering (MVC)* to find the smallest set of nodes that every edge is connected to at least one nodes in the set, and *max independent set (MIS)* to find the largest set where any two vertices in the set are not adjacent. Their objectives (Eq. 1) and relaxations (Eq. 2) are listed in Table 2. For the detailed derivation, see Appendix C.

### 5.1 SETTINGS

**Datasets:** We conduct experiments on the MC and MVC problems over three real datasets Twitter (Leskovec & Krevl, 2014), COLLAB and IMDB (Yanardag & Vishwanathan, 2015) and two synthetic datasets with 200 and 500 nodes generated by the RB model (Xu, 2007). We name them RB200 and RB500, respectively. We make RB200 and RB500 extremely hard by setting a small hyper-parameter $\rho = 0.25$ of the RB model (Xu, 2007). The difficulty-$\rho$ relationship on the MVC problem with 500 vertices is shown in Fig. 3, where the models are pre-trained on the RB graphs with uniformly sampled $\rho \in [0.3, 1.0]$ and tested on different RB graphs generated with single $\rho$'s. We keep all the other hyper-parameters the same. As $\rho$ increases, Meta-EGN and Gurobi9.5 all tend to achieve better performance. Meta-EGN could outperform Gurobi9.5 in hard instances $\rho \in (0, 0.55]$ while remaining a gap on the easy ones. To verify performances for data-scale generalization, we also generate large-graph datasets RB1000, RB2000, and RB5000 with $\rho = 0.25$. As for the MIS problem, random-regular graphs (RRGs) are often used as benchmarks because they are challenging.

Table 3: ApR (time: second/graph) on the MC problem. ApR is the larger the better. 'report' denotes the reported performance in Karalias & Loukas (2020), 're-impl' denotes re-implementation, 'f-t' stands for fine-tune. Pareto-optimal results are in bold.

|  | Twitter | COLLAB | IMDB | RB 200 | RB 500 |
|---|---|---|---|---|---|
| EGN (report) | $0.924 \pm 0.133$ (0.17) | $0.982 \pm 0.063$ (0.10) | 1.000 (0.08) | - | - |
| EGN (re-impl) | $0.926 \pm 0.113$(0.17) | $0.982 \pm 0.069$ (0.10) | 1.000 (0.08) | $0.882 \pm 0.156$ (0.13) | $0.852 \pm 0.217$ (0.33) |
| EGN (re-impl) f-t | $0.949 \pm 0.102$(0.49) | $0.986 \pm 0.060$(0.27) | 1.000 (0.20) | $0.926 \pm 0.158$ (0.49) | $0.899 \pm 0.469$ (0.91) |
| RUN-CSP | $0.909 \pm 0.145$ (0.21) | $0.912 \pm 0.188$ (0.14) | $0.823 \pm 0.191$ (0.11) | $0.926 \pm 0.302$ (0.47) | $0.854 \pm 0.870$ (2.09) |
| Meta-EGN | **$0.976 \pm 0.048$(0.17)** | $0.988 \pm 0.059$ (0.10) | 1.000 (0.08) | **$0.930 \pm 0.143$ (0.13)** | **$0.875 \pm 0.213$ (0.33)** |
| Meta-EGN f-t | $0.990 \pm 0.028$(0.49) | $0.993 \pm 0.038$ (0.27) | 1.000 (0.20) | **$0.976 \pm 0.139$ (0.49)** | **$0.917 \pm 0.208$ (0.91)** |
| Toenshoff-Greedy | **$0.917 \pm 0.126$ (0.08)** | $0.969 \pm 0.087$ (0.06) | $0.987 \pm 0.050$ (1e-3) | $0.841 \pm 0.139$ (0.23) | $0.846 \pm 0.197$ (3.83) |
| Gurobi9.5 ($\leq$0.20s) | $0.737 \pm 0.267$ (0.17) | **$0.871 \pm 0.242$ (0.04)** | **1.000 (1e-3)** | - | - |
| Gurobi9.5 ($\leq$1.00s) | **1.000 (0.37)** | $0.979 \pm 0.117$ (0.06) | 1.000 (1e-3) | $0.898 \pm 0.092$ (0.36) | - |
| Gurobi9.5 ($\leq$2.50s) | 1.000 (0.37) | $0.997 \pm 0.036$ (0.06) | 1.000 (1e-3) | **$0.987 \pm 0.039$ (1.65)** | $0.778 \pm 0.209$ (1.68) |
| Gurobi9.5 ($\leq$4.00s) | 1.000 (0.37) | **1.000 (0.06)** | 1.000 (1e-3) | **1.000 (2.61)** | $0.820 \pm 0.211$ (2.86) |

Table 4: ApR (time: second/graph) on the MVC problem. ApR is the smaller the better.. 'f-t' stands for one-step fine-tune. Pareto-optimal results are in bold.

|  | Twitter | COLLAB | IMDB | RB 200 | RB 500 |
|---|---|---|---|---|---|
| EGN | $1.033 \pm 0.023$(0.29) | $1.013 \pm 0.022$ (0.15) | 1.000 (0.08) | $1.031 \pm 0.004$ (0.26) | $1.021 \pm 0.002$ (0.48) |
| EGN f-t | $1.028 \pm 0.021$(0.80) | $1.008 \pm 0.015$ (0.38) | 1.000 (0.32) | $1.030 \pm 0.005$ (0.80) | $1.021 \pm 0.002$ (1.59) |
| RUN-CSP | $1.180 \pm 0.435$ (0.16) | $1.208 \pm 0.198$ (0.19) | $1.188 \pm 0.178$ (0.08) | $1.124 \pm 0.001$ (0.28) | $1.062 \pm 0.005$ (1.65) |
| Meta-EGN | $1.019 \pm 0.017$(0.29) | $1.003 \pm 0.010$ (0.15) | 1.000 (0.08) | **$1.028 \pm 0.005$ (0.26)** | **$1.016 \pm 0.002$ (0.48)** |
| Meta-EGN f-t | $1.017 \pm 0.017$(0.80) | $1.002 \pm 0.010$ (0.38) | 1.000 (0.32) | $1.027 \pm 0.006$ (0.80) | $1.016 \pm 0.002$ (1.59) |
| Greedy | $1.014 \pm 0.014$ (1.95) | $1.209 \pm 0.198$ (1.79) | $1.180 \pm 0.077$ (0.02) | $1.124 \pm 0.002$ (5.02) | $1.062 \pm 0.005$ (15.59) |
| Gurobi9.5 ($\leq$0.25s) | **$1.028 \pm 0.054$ (0.09)** | $1.002 \pm 0.010$ (0.10) | **1.000 (0.01)** | - | - |
| Gurobi9.5 ($\leq$0.50s) | $1+1e-3 \pm 0.001$ (0.13) | **1.000 (0.10)** | 1.000 (0.01) | - | - |
| Gurobi9.5 ($\leq$1.00s) | **1.000 (0.13)** | 1.000 (0.10) | 1.000 (0.01) | **$1.011 \pm 0.003$ (0.63)** | $1.019 \pm 0.003$ (0.69) |
| Gurobi9.5 ($\leq$2.00s) | 1.000 (0.13) | 1.000 (0.10) | 1.000 (0.01) | **$1.008 \pm 0.002$ (1.16)** | $1.019 \pm 0.003$ (1.68) |

Our experiments also use RRGs by following the settings of (Schuetz et al., 2022) with the node number ranging from $10^2$ to $10^5$ and the node degree selected from the set $\mathcal{D} = \{3, 7, 10, 20\}$. Here, node degree equaling 20 is the hardest setting (Angelini & Ricci-Tersenghi, 2022). A summary of these datasets is in Table. 10 in Appendix.

**Data Splitting & The Evaluation Metric:** For the real datasets, training/validation/test instances are randomly divided with the ratio of 7:1:2; For RB200 and RB500, $2000/100/100$ graphs are generated for training/validation/test instances; For RB1000, RB2000, RB5000, we generate 100 test instances. As to RRG datasets, the training set contains 3000 RRGs, of which each has 1000 nodes and the node degree is uniformly sampled from $\mathcal{D}$. We generate 15 graphs for validation and 20 graphs for each node degree configuration in $\mathcal{D}$ for test. Our evaluation metric uses the approximation rate (ApR). All results are summarized based on 5 times independent experiments with different random seeds.

**Baselines:** Our baselines include unsupervised learning methods, heuristics, and traditional CO solvers. For the MC and MVC problems, we take our direct baseline EGN (Karalias & Loukas, 2020), and also take RUN-CSP (Toenshoff et al., 2021) as another baseline. We do not consider other learning-based methods because they generally perform worse than EGN (Karalias & Loukas, 2020). As to the heuristics, we use greedy algorithms as heuristic baselines. For traditional CO solvers, we compare against the best commercial CO problem solver Gurobi9.5 (Gurobi Optimization, 2022) via converting the problems into integer programming form. We track the time $t$ that the models use from the start of inferring to the end of rounding to output feasible solutions. We set this time $t$ as the time budget of Gurobi9.5 for purely solving the integer programming, and list the actual time usage of Gurobi9.5 which includes pre-processing plus $t$. As to the MIS problem, we take PI-GNN (Schuetz et al., 2022) and EGN Karalias & Loukas (2020) as the learning-based baselines. We take the random greedy algorithm (RGA) and degree-based greedy algorithm (DGA) as introduced in Angelini & Ricci-Tersenghi (2019) as the heuristic baselines. When we consider fine-tuning EGN and Meta-EGN over a test instance, we use 1-step gradient descent as fine-tuning.

**Implementation:** For the MC and MVC problems, we use 4-layer GIN (Xu et al., 2019) as the backbone network for both meta-EGN and EGN. We use 1e-3 as both the outer learning rate ($\gamma$) of Meta-EGN and the learning rate of EGN. Here, the backbone and the learning rate are the same as those in (Karalias & Loukas, 2020). For the MIS problem, we use 6-layer GIN. The outer learning rate ($\gamma$) of Meta-EGN and the learning rate of EGN are set as 1e-4. The inner learning rate ($\alpha$) of Meta-EGN is always set as 5e-5. We run all experiments by using a Xeon(R) Gold 6248 CPU with

Table 5: Scale generalization on the MC and MVC. ApR is the larger the better for MC while the smaller the better for MVC. All the models are trained on RB500 training data. 'Fast/Medium/Accurate' denotes GNNs (without fine-tuning) using 1/4/8 random single node seed(s) per testing instance. 'Fine-tuning' use 1-step Fine-tuning the best trial among the 8 node seed(s). 'Gap' represents the averaged gap defined as $c \times$ (# of nodes in the optimal solution - # of nodes by the given method) where $c = 1$ for MC and $c = -1$ for MVC. 'Rank' is the average rank of solutions among the three methods. Optimal solutions are generated via Gurobi9.5 with a time limit 3000 seconds. Approximation rate for MC larger than 1, highlighted by [*], indicates the model outperforms Gurobi9.5 solver with 3000s time budget. Pareto-optimal results are in bold.

| | Dataset | Method | Fast (1) | | | Medium (4) | | | Accurate (8) | | | Fine-tune | | |
|---|---|---|---|---|---|---|---|---|---|---|---|---|---|---|
| | | | ApR(s/g) | Gap | Rank | ApR(s/g) | Gap | Rank | ApR(s/g) | Gap | Rank | ApR(s/g) | Gap | Rank |
| MC | RB1000 | EGN | 0.646±0.282(0.05) | 11.48 | 2.406 | 0.843±0.229(0.17) | 6.47 | 2.237 | 0.909±0.205(0.33) | 4.86 | 2.025 | 0.963±0.186(0.98) | 4.13 | 1.693 |
| | | Meta-EGN | 0.769±0.276(0.05) | 8.57 | 1.943 | **0.938±0.196(0.17)** | **4.61** | **1.543** | **0.940±0.205(0.33)** | **4.97** | **1.581** | **0.974±0.195(0.98)** | **4.01** | **1.625** |
| | | Gurobi9.5 | **0.885±0.197(6.11)** | **5.08** | **1.650** | 0.885±0.197(6.18) | 5.08 | 2.218 | 0.885±0.197(6.48) | 5.08 | 2.393 | 0.885±0.197(7.01) | 5.08 | 2.681 |
| | RB2000 | EGN | 0.679±0.290(0.10) | 12.38 | 2.408 | 0.896±0.184(0.29) | 5.23 | 2.136 | 0.945±0.160(0.58) | 3.81 | 2.208 | 0.971±0.154(2.03) | 3.61 | 1.983 |
| | | Meta-EGN | 0.807±0.114(0.10) | 8.13 | 1.991 | **0.978±0.157(0.29)** | **3.48** | **1.591** | **0.995±0.146(0.58)** | **1.28** | **1.466** | **1.011±0.134(2.03)[*]** | **0.55** | **1.483** |
| | | Gurobi9.5 | **0.951±0.145(24.14)** | **3.01** | **1.600** | 0.951±0.145(24.56) | 3.01 | 2.091 | 0.951±0.145(25.01) | 3.01 | 2.325 | 0.951±0.145(25.66) | 3.01 | 2.533 |
| | RB5000 | EGN | 0.960±0.159(0.33) | 2.42 | 2.130 | 1.020±0.139(1.02)[*] | -1.26 | 2.060 | 1.027±0.140(2.50)[*] | -1.68 | 1.980 | 1.047±0.188(9.66)[*] | -2.86 | 1.970 |
| | | Meta-EGN | **1.028±0.138(0.33)[*]** | **-1.62** | **1.820** | **1.068±0.233(1.02)[*]** | **-4.02** | **1.820** | **1.072±0.234(2.50)[*]** | **-4.42** | **1.790** | **1.077±0.233(9.66)[*]** | **-4.72** | **1.710** |
| | | Gurobi9.5 | 1.000(201.55) | 0.00 | 2.050 | 1.000(202.36) | 0.00 | 2.120 | 1.000(205.64) | 0.00 | 2.230 | 1.000(214.35) | 0.00 | 2.320 |
| | | Gurobi9.5 | 1.000(3000) | 0.00 | - | 1.000(3000) | 0.00 | - | 1.000(3000) | 0.00 | - | 1.000(3000) | 0.00 | - |
| MVC | RB1000 | EGN | 1.0161±0.0048(0.20) | 16.46 | 2.250 | 1.0135±0.0013(0.72) | 13.73 | 1.920 | 1.0138±0.0013(1.37) | 13.29 | 1.860 | 1.0138±0.0013(3.05) | 13.73 | 1.960 |
| | | Meta-EGN | 1.0145±0.0016(0.20) | 14.81 | 1.935 | **0.0131±0.0012(0.72)** | **13.40** | **1.700** | **1.0125±0.0012(1.37)** | **12.75** | **1.545** | **1.0124±0.0012(3.05)** | **12.69** | **1.455** |
| | | Gurobi9.5 | **1.0143±0.0018(1.92)** | **14.58** | **1.835** | 1.0143±0.0018(2.58) | 14.58 | 2.380 | 1.0143±0.0018(3.08) | 14.58 | 2.585 | 1.0143±0.0018(4.96) | 14.58 | 2.585 |
| | RB2000 | EGN | 1.0114±0.0026(0.34) | 22.02 | 2.350 | 1.0096±0.0008(1.32) | 18.57 | 1.765 | 1.0094±0.0007(2.69) | 18.17 | 1.765 | 1.0093±0.0007(6.27) | 17.98 | 1.890 |
| | | Meta-EGN | **0.0103±0.0015(0.34)** | **19.94** | **1.740** | **1.0095±0.0008(1.32)** | **18.41** | **1.635** | **1.0092±0.0007(2.69)** | **17.82** | **1.510** | **1.0090±0.0006(6.27)** | **17.38** | **1.360** |
| | | Gurobi9.5 | 1.0104±0.0010(5.63) | 20.18 | 2.910 | 1.0104±0.0010(6.65) | 20.18 | 2.600 | 1.0104±0.0010(8.04) | 20.18 | 2.725 | 1.0104±0.0010(13.24) | 20.18 | 2.750 |
| | RB5000 | EGN | 1.0071±0.0014(1.01) | 34.19 | 2.170 | 1.0064±0.0004(3.99) | 30.83 | 1.985 | 1.0062±0.0004(7.95) | 29.87 | 1.865 | 1.0062±0.0004(18.41) | 29.68 | 1.960 |
| | | Meta-EGN | **1.0067±0.0005(1.01)** | **32.51** | **2.045** | **1.0062±0.0005(3.99)** | **29.96** | **1.600** | **1.0061±0.0004(7.95)** | **29.44** | **1.555** | **1.0060±0.0003(18.41)** | **29.15** | **1.470** |
| | | Gurobi9.5 | 1.0066±0.0006(24.60) | 31.88 | 1.785 | 1.0066±0.0006(28.72) | 31.88 | 2.415 | 1.0066±0.0006(32.16) | 31.88 | 2.580 | 1.0066±0.0006(42.62) | 31.88 | 2.570 |

Table 6: Generalization from Twitter to RB2000 on the MC and MVC. Pareto-optimal results are in bold.

| Method | MC (Approximation Rate ↑ (time)) | | | | MVC (Approximation Rate ↓ (time)) | | | |
|---|---|---|---|---|---|---|---|---|
| | Fast (1) | Medium (4) | Accurate (8) | Fine-tune | Fast (1) | Medium (4) | Accurate (8) | Fine-tune |
| EGN | 0.594±0.210(0.07) | 0.788±0.201(0.16) | 0.819±0.195(0.29) | 0.831±0.192(0.89) | 1.055±0.005(0.11) | 1.053±0.004(0.37) | 1.052±0.004(0.48) | 1.050±0.004(1.59) |
| Meta-EGN | **0.690±0.201(0.07)** | **0.793±0.197(0.16)** | **0.833±0.193(0.29)** | **0.876±0.182(0.89)** | 1.036±0.005(0.11) | 1.030±0.003(0.37) | 1.029±0.002(0.48) | 1.021±0.003(1.59) |
| Gurobi9.5 | 0.663±0.188(2.92) | 0.663±0.188(2.92) | 0.669±0.191(3.08) | 0.742±0.213(3.88) | **1.019±0.003(1.12)** | **1.019±0.003(1.30)** | **1.019±0.003(1.35)** | **1.017±0.002(2.40)** |

26 threads and a Quadro RTX 6000 GPU. All codes run on the PyTorch platform (Paszke et al., 2019). For more details, see Appendix. C.

**Overcoming the limited expressive power of GNNs:** GNNs are known with limited expressive power (Xu et al., 2019; Morris et al., 2019). Specifically, over RRGs, the GIN backbone will associate each node with the same representation, unless node representations are initialized not equally. To keep a fair comparison, for the MC and MVC problem, we follow Karalias & Loukas (2020) and adopt the initialization based on a single random node seed (one selected node is initialized as 1, others as 0). We use 8 single random node seeds for EGN and Meta-EGN in the experiments of Sec. 5.2 and report the best among the 8 trials. We try different numbers of random node seeds in the experiments of Sec. 5.3. For the large-scale MIS problem studied in Sec. 5.4, we find such single node initialization is too local to generate valid global solutions. So, we adopt initialization based on the solutions of greedy algorithms DGA (for Figs. 1,4.2) and RGA (for Fig. 4). Then, EGN and Meta-EGN can be viewed as learning heuristics to improve the greedy solutions. Note that learning heuristics to tune these solutions is non-trivial (Andrade et al., 2012; Rahman & Virag, 2017).

## 5.2 META-EGN BOOSTS THE PERFORMANCE WITHOUT DISTRIBUTION SHIFTS

We first compare the performances of different methods when the datasets used for training and testing are from the same distribution. Table 3 and Table 4 show the results for the MC problem and the MVC problem respectively. In both problems and across the five datasets, Meta-EGN significantly outperforms EGN and RUN-CSP, both before and after the fine-tuning step. In comparison with the traditional CO solvers, Meta-EGN narrows the gap from Gurobi9.5 on those real small graphs. For RB graphs, Meta-EGN outperforms Gurobi9.5 on RB500 for both the MC and MVC problems.

We notice that both EGN and Meta-EGN perform generally well on the MC problem while not as competitive on the MVC problem. This results from the initialization of GNN inputs. The MC problem outputs clusters that are more local while MVC asks for global assignments, which makes such single-seed-based initialization less fit for the MVC problem.

## 5.3 META-EGN BOOSTS THE PERFORMANCE WITH DISTRIBUTION SHIFTS

**Problem Scale Shift:** Here, we use large-scale RB graphs of 1000-5000 nodes to test EGN and Meta-EGN that is trained based on RB500. Table 5 shows the results. Both methods show good generalization while Meta-EGN is always better. As the scale increases, Meta-EGN outperforms Gurobi9.5. For example, it takes Meta-EGN with 4 random initializations only 1.02s to beat

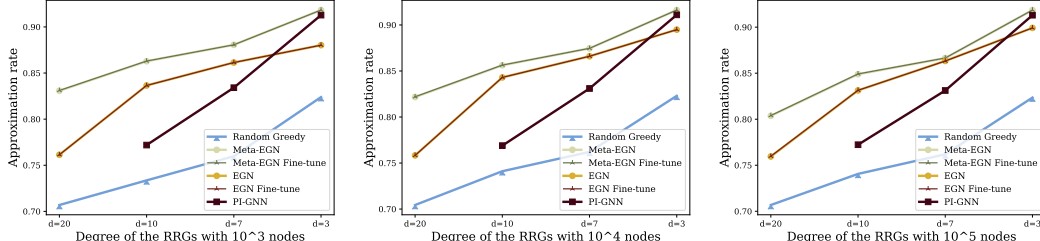

Figure 4: ApRs in the MIS problem on RRGs. Meta-EGN and EGN are both trained with the output of Random Greedy Algorithm (RGA) as initialization.

Gurobi9.5 that runs for 3000 seconds on RB5000 dataset in the MC problem. Moreover, Meta-EGN can even outperform Gurobi9.5 on the MVC problem when the problem scale becomes large.

**Real-Synthetic Distribution Shift:** Here, we train EGN and Meta-EGN on Twitter and test them on RB500. Table 6 shows the results. Compare Table 6 with Tables 3,4. We observe better generalization performance of Meta-EGN compared to EGN. For example, for the MC, Meta-EGN has almost the same performance whether there is a dataset shift or not (0.833 v.s. 0.834 before fine-tuning, 0.876 v.s. 0.878 after fine-tuning) while EGN has a bigger gap in performance when there is a shift (0.819 v.s. 0.829 before fine-tuning, 0.831 v.s. 0.864 after fine-tuning). For the MVC, although the performance drop of Meta-EGN is larger, such a drop is still much smaller than that of EGN.

## 5.4 MAX INDEPENDENT SET: A RESPONSE TO (ANGELINI & RICCI-TERSENGHI, 2022)

For the MIS problem on large-scale RRGs, Angelini & Ricci-Tersenghi (2022) have recently posted a concern on learning-based methods by arguing that PI-GNN in Schuetz et al. (2022) could not achieve comparable results with the heuristic algorithm DGA (Angelini & Ricci-Tersenghi, 2019). We see the reason comes from improper usage of learning-based methods in Schuetz et al. (2022) as stated in Sec. 1: 1) PI-GNN is trained directly on each single testing instance without learning from the training dataset that contains varies graphs, which is likely to be trapped into the local optima; 2) GNN generally suffers from a node ambiguity issue on RRGs. To resolve the problem, we utilize the outputs of DGA and RGA as the initialization of GNN inputs (EGN, Meta-EGN) and expect to learn heuristics from historical data to further tune the solutions given by the greedy algorithms. We train GNN models on RRGs with 1000 nodes with node degrees randomly sampled from $3, 7, 10, 20$, and test on larger RRGs (up to $10^5$ nodes). Experiments show that Meta-EGN can further improve DGA (in Fig. 1) and RGA (in Fig. 4), while EGN fails to better tune DGA. Note that here EGN and Meta-EGN adopt the exactly same backbones. We attribute the improvement to meta-learning-based training as adopted by Meta-EGN. See Table 7 in Appendix for more details of the numerical improvement by Meta-EGN. We also see in these cases, one-step fine-tuning does not contribute much to the performance of EGN or Meta-EGN, indicating the model before fine-tuning has been very close to a local minimum.

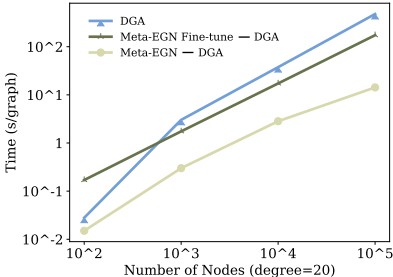

Figure 5: Time cost v.s. Graph Scales.

We also check the extra time cost by running Meta-EGN to improve DGA solutions in Fig. 5 and Fig. 6 in Appendix. The extra time cost is just 1% (without fine-tuning) - 30% (with fine-tuning) of the time cost of DGA. In theory, the extra time cost without fine-tuning should be $O(|E|)$ for GNN inference plus $O(|V|)$ for rounding, which is in the same order as DGA, while the GNN parallel inference substantially reduces the time.

## 6 CONCLUSION

This work proposes an unsupervised learning framework Meta-EGN with the goal of optimizing NNs towards instance-wise good solutions to CO problems. Meta-EGN leverages MAML to achieve the goal. Meta-EGN views each training instance as a separate task and learns a good initialization for all these tasks. Meta-EGN significantly improves the performance of its baseline and has shown good generalization when the data used for training and testing has different scales or distributions. In addition, Meta-EGN can learn to improve the greedy heuristics while paying almost no extra time cost in the problem of maximum independent set on large-scale random regular graphs.

## 7 ACKNOWLEDGEMENT

We would like to express our deepest appreciation to Dr. Tianyi Chen for the insightful discussion on the meta-learning framework from a theoretical aspect and Dr. Ruqi Zhang for the constructive advice on the fine-tuning strategies. We would also like to extend our deepest gratitude to Dr. Hanjun Dai and Dr. Jialin Liu for sharing their invaluable insights into the general ideas of learning for combinatorial optimization. Also many thanks to our funding, H. Wang and P. Li are partially supported by 2021 JPMorgan Faculty Award and the NSF award OAC-2117997.

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

## A    PROOF OF THEOREM 1

We first prove Theorem 1, then we specify the value of $\alpha$ to obtain Theorem 2 as a specific case of Theorem 1. The proof of Theorem 1 is divided into two parts.

In part 1, we prove that if $l(\theta; G) < \beta + \triangle$ (even if $l(\theta; G) \geq \beta$), for any $\alpha \in (0, 2/L)$ Meta-GNN with one-step finetuning outputs a feasible solution $X$ of good quality $f(X; G) \leq l(\theta; G) - \triangle$. Here, $\triangle = \|\nabla_\theta l(\theta; G)\|\epsilon + \frac{1}{2L\alpha^2 - 4\alpha}\epsilon^2$ if $\epsilon < \alpha\|\nabla_\theta l(\theta; G)\|$ or $\triangle = (\alpha - \frac{L\alpha^2}{2})\|\nabla_\theta l(\theta; G)\|^2$ o.w..

In part 2, we prove that once Meta-EGN achieves the loss value $l(\theta'; G)$ after the one-step finetuning, the rounding process would output a feasible $X$ whose objective satisfies $f(X; G) \leq l(\theta'; G)$.

**Part 1:** We could get

$$
\begin{aligned}
l(\theta'; G) &\overset{(a)}{\leq} l(\theta; G) + \nabla_\theta l(\theta; G)(\theta' - \theta) + \frac{1}{2}L\|\theta' - \theta\|_2^2 \\
&\overset{(b)}{=} l(\theta; G) + \frac{1}{2}L\|\theta' - \theta\|_2^2 - \alpha\|\nabla_\theta l(\theta; G)\|_2^2 \\
&= l(\theta; G) + (\frac{L\alpha^2}{2} - \alpha)\|\nabla_\theta l(\theta; G)\|_2^2,
\end{aligned}
\tag{6}
$$

where (a) is due to the local L-smoothness of $l(\cdot; G)$, (b) is due to the definition of one-step finetuning $\theta' = \theta - \alpha \nabla_\theta l(\theta; G)$.

If $\epsilon < \alpha \|\nabla_\theta l(\theta; G)\|$:

Let $\triangle = \|\nabla_\theta l(\theta; G)\|\epsilon + \frac{1}{2L\alpha^2 - 4\alpha}\epsilon^2$, we have:

$$\min_\epsilon -\triangle = \min_\epsilon -\frac{1}{2L\alpha^2 - 4\alpha}\epsilon^2 - \|\nabla_\theta l(\theta; G)\|\epsilon = (\frac{L\alpha^2}{2} - \alpha)\|\nabla_\theta l(\theta; G)\|^2, \tag{7}$$

thus

$$l(\theta'; G) \leq l(\theta; G) - \triangle. \tag{8}$$

If $\epsilon \geq \alpha \|\nabla_\theta l(\theta; G)\|$:

Let $\triangle = (\alpha - \frac{L\alpha^2}{2})\|\nabla_\theta l(\theta; G)\|^2$, we would directly have:

$$l(\theta'; G) \leq l(\theta; G) - \triangle. \tag{9}$$

By this, we finish the first part of the proof for Theorem 1.

**Part 2:** The proof in this part follows the rounding analysis in Wang et al. (2022). Consider the rounding procedure from continuous space $\bar{X} = \mathcal{A}_\theta(G), \bar{X} \in [0,1]^n$ into the discrete feasible solution $X \in \{0,1\}^n$. Let $\bar{X}_i, X_i, i = \{0, 1, ..., n\}$ denote their entries. W.l.o.g, suppose the rounding order is from 1 to $n$ and we have finished the rounding before the $t$-th node, we now analyze the rounding of $t$-th node:

$$f_r([X_1, ..., X_{t-1}, \bar{X}_t, \bar{X}_{t+1}, ..., \bar{X}_n]; G) + \beta g_r([X_1, ..., X_{t-1}, \bar{X}_t, \bar{X}_{t+1}, ..., \bar{X}_n]; G)$$

$$\overset{(d)}{\geq} \bar{X}_t(f_r([X_1, ..., X_{t-1}, 1, \bar{X}_{t+1}, ...\bar{X}_n]; G) + \beta g_r([X_1, ..., X_{t-1}, 1, \bar{X}_{t+1}, ..., \bar{X}_n]; G))$$

$$+ (1 - \bar{X}_t)(f_r([X_1, ..., X_{t-1}, 0, \bar{X}_{t+1}, ..., \bar{X}_n]; G) + \beta g_r([X_1, ..., X_{t-1}, 0, \bar{X}_{t+1}, ..., \bar{X}_n]; G))$$

$$\geq \bar{X}_t(\min_{j_t = \{0,1\}} f_r([X_1, ..., X_{t-1}, j_t, \bar{X}_{t+1}, ..., \bar{X}_n]; G) + \beta g_r([X_1, ..., X_{t-1}, j_t, \bar{X}_{t+1}, ..., \bar{X}_n]; G))$$

$$+ (1 - \bar{X}_t)(\min_{j_t = \{0,1\}} f_r([X_1, ..., X_{t-1}, j_t, \bar{X}_{t+1}, ..., \bar{X}_n]; G)$$

$$+ \beta g_r([X_1, ..., X_{t-1}, j_t, \bar{X}_{t+1}, ..., \bar{X}_n]; G))$$

$$\overset{(e)}{=} f_r([X_1, ..., X_{t-1}, X_t, \bar{X}_t, ..., \bar{X}_n]; G) + \beta g_r([X_1, ..., X_{t-1}, X_t, \bar{X}_t, ..., \bar{X}_n]; G)$$

$$\tag{10}$$

where (d) is due to $l_r(\theta; G)$'s entry-wise concavity w.r.t $\bar{X}$ and Jensen's inequality, (e) is due to $X_t = \arg\min_{j=0,1} f_r(X_1, ..., X_{t-1}, t, \bar{X}_{t+1}, ..., \bar{X}_n) + \beta g_r(X_1, ..., X_{t-1}, t, \bar{X}_{t+1}, ..., \bar{X}_n)$ (the definition of our rounding process). The loss value is monotonically non-increasing through the whole rounding process according to the equation above, thus we could get:

$$l(\theta') \geq f(X; G) + \beta g(X; G) \tag{11}$$

By this, we finish the proof of the second part.

### A.1 A SPECIFIC CASE

Note that in the first part of the proof above, if we specify the value of $\alpha$ as $\frac{1}{L}$ in equation (6), we could have:

$$l(\theta'; G) \leq l(\theta; G) - \frac{\|\nabla_\theta l(\theta; G)\|_2^2}{2L} \tag{12}$$

If $\epsilon < \frac{1}{L}\|\nabla_\theta l(\theta; G)\|$:

Let $\triangle = \|\nabla_\theta l(\theta; G)\|\epsilon - \frac{L}{2}\epsilon^2$, we have:

$$\min_\epsilon -\triangle = \min_\epsilon \frac{L}{2}\epsilon^2 - \|\nabla_\theta l(\theta; G)\|\epsilon = -\frac{\|\nabla_\theta l(\theta; G)\|^2}{2L}, \tag{13}$$

thus

$$l(\theta'; G) \leq l(\theta; G) - \triangle. \tag{14}$$

If $\epsilon \geq \frac{1}{L}\|\nabla_\theta l(\theta; G)\|$:

Let $\triangle = \frac{1}{2L}\|\nabla_\theta l(\theta; G)\|^2$, we would directly have:

$$l(\theta'; G) \leq l(\theta; G) - \triangle. \tag{15}$$

By this, we obtain Theorem 2, a specific case of Theorem 1 as follows:

**Theorem 2** (A Specific case of Theorem 1). *Suppose the relaxations $f_r$ and $g_r$ are entry-wise concave as required in (Wang et al., 2022). Let $\theta$ denote the learned parameter after training. Given a test instance $G$, suppose locally $l(\cdot; G)$ is L-smooth at $\theta$, i.e., $\|\nabla_{\theta'} l(\theta'; G) - \nabla_\theta l(\theta; G)\| \leq L\|\theta' - \theta\|$ for all $\theta'$ that satisfies $\|\theta' - \theta\| \leq \epsilon$. Then, if $\underline{l(\theta; G) < \beta + \triangle}$ (even if $l(\theta; G) \geq \beta$), there exists $\alpha$ such that Meta-GNN with one-step finetuning outputs a feasible solution $X$ of good quality $f(X; G) \leq l(\theta; G) - \triangle$. Here, $\triangle = \|\nabla_\theta l(\theta; G)\|\epsilon - \frac{L}{2}\epsilon^2$ if $\epsilon < \frac{1}{L}\|\nabla_\theta l(\theta; G)\|$ or $\triangle = \frac{1}{2L}\|\nabla_\theta l(\theta; G)\|^2$ o.w..*

# B    SUPPLEMENTARY EXPERIMENT RESULTS

## B.1    SUPPLEMENTARY TIME COST V.S. GRAPH SCALE IN THE MIS

we show the degree $3, 7, 10$ in the following Fig. 6. They show the same time-cost vs scale relation as that in Fig. 5. The extra time cost of GNN is $O(|E|)$ for inference plus $O(|V|)$ for rounding, which is in the same order of DGA.

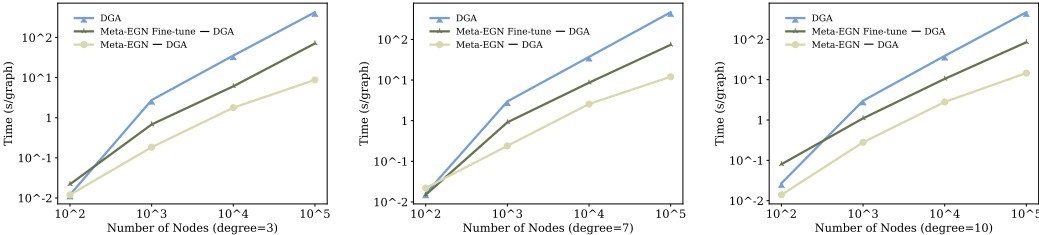

Figure 6: Time cost v.s. Graph Scales on degree $3, 7, 10$

## B.2    HOW MUCH DOES META-EGN MODIFY DGA AND RGA HEURISTICS IN THE MIS

We display the average approximation rate improvement and the average node number increase by Meta-EGN over DGA and RGA in Table. 7.

Table 7: Improvement of Meta-EGN over DGA and RGA in the MIS on RRGs, 'Imp in ApR' denotes the average improvement in approximation rate, and 'Imp in #Node' denotes the average number of nodes that Meta-EGN could find more than the heuristics.

| | Scale/Degree | 3 | | 7 | | 10 | | 20 | |
|---|---|---|---|---|---|---|---|---|---|
| | | Imp in ApR | Imp in #Node | Imp in ApR | Imp in #Node | Imp in ApR | Imp in #Node | Imp in ApR | Imp in #Node |
| | $10^3$ | 0.0043 | 1.950 | 0.0060 | 2.014 | 0.0044 | 1.254 | 0.0084 | 1.657 |
| Meta-EGN improves DGA by | $10^4$ | 0.0050 | 22.768 | 0.0062 | 20.811 | 0.0067 | 19.109 | 0.0079 | 15.588 |
| | $10^5$ | 0.0032 | 145.718 | 0.0045 | 151.051 | 0.0051 | 145.69 | 0.0050 | 98.660 |
| | $10^3$ | 0.0944 | 42.986 | 0.1208 | 40.549 | 0.1292 | 36.849 | 0.1239 | 24.447 |
| Meta-EGN improves RGA by | $10^4$ | 0.0932 | 424.404 | 0.1125 | 377.628 | 0.1151 | 328.276 | 0.1173 | 231.456 |
| | $10^5$ | 0.0871 | 3966.272 | 0.1045 | 3507.751 | 0.1083 | 3088.824 | 0.0969 | 1912.030 |

## B.3    TRAINING THE MODELS ON SUBSETS OF THE TRAINING DATA

We display the average approximation rates of the models that are only trained on subsets of the original training data in the max clique problem on Twitter. The training dataset is randomly sampled from the original training dataset and the testing dataset remains the same as that in Table. 3. Both methods have worse performance as the number of training instances reduces, while Meta-EGN only has a $0.6\%$ performance decrease from the full-size training dataset with 695 samples to the training subset with only 64 instances. In contrast, EGN decreases its performance by $1.7\%$.

Table 8: The approximation rate of the max clique problem on Twitter. Models are only trained on subsets of the dataset, 'training subset' denotes the number of instances in the training data.

| training subset | 64 | 128 | 256 | 512 | Full (665) |
|---|---|---|---|---|---|
| EGN | 0.909±0.122 | 0.911±0.118 | 0.914±0.118 | 0.922±0.115 | 0.926±0.113 |
| Meta-EGN | 0.970±0.058 | 0.973±0.055 | 0.975±0.055 | 0.975±0.051 | 0.976±0.048 |

### B.4 TRAINING THE MODELS ON RRGs WITH SINGLE DEGREES IN THE MIS

We train the EGN and Meta-EGN models on RRGs with only 3 or 20 degrees and test them on RRGs with the rest of degrees from $\{3, 7, 10, 20\}$. Models take the output of DGA as the initialization graph node feature. We show the performance of both the models without fine-tuning in Fig. 7. When only trained on RRGs with degree 3 (See the left two figures in Fig. 7), both the models could not generalize well, as neither of them could outperform the initialization input of DGA. Note that Meta-EGN still achieves better performance than EGN in this case. As to the models only trained on RRGs with degree 20 (See the right two figures in Fig. 7), we observe that both the model have relatively good generalization ability across different degrees, yet Meta-EGN could still marginally outperform EGN in this case. We attribute this phenomenon to the fact that solving the MIS on RRGs with degree 20 is much more complicated than those with degree 3 and thus may contain adequate heuristics for solving RRGs with lower degrees.

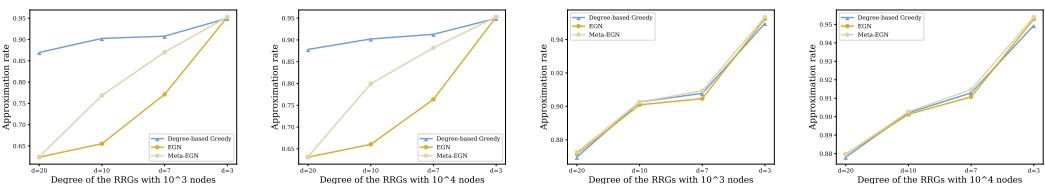

Figure 7: The left two figures show the ApRs on RRGs with $10^3$ and $10^4$ nodes of the models trained only on RRGs with degree 3. The right two figures show the ApRs on RRGs with $10^3$ and $10^4$ nodes of the models trained only on RRGs with degree 20.

### B.5 COMPARISON ON THE TRAINING TIME OF THE MODELS

We display the wall clock training time for the two methods to converge in Table. 9 (from start to the best epoch on validation set). We observe that Meta-EGN generally takes two to three times to converge compared with EGN, but their training time cost basically remains on the same order of magnitude.

Table 9: The wall clock training time to convergence of EGN and Meta-EGN in different problems.

| Dataset/Time (min:second) | MC | | | MVC | | | MIS |
|---|---|---|---|---|---|---|---|
| | Twitter | RB200 | RB500 | Twitter | RB200 | RB500 | RRGs |
| EGN | 46:50 | 104:37 | 282:57 | 100:58 | 83:27 | 128:39 | 733:02 |
| Meta-EGN | 101:55 | 210:04 | 609:47 | 276:38 | 168:25 | 282:15 | 1088:55 |

## C SUPPLEMENTARY IMPLEMENTATION DETAILS

### C.1 EXPERIMENT DETAILS

All the codes run on the PyTorch platform 1.9.0 (Paszke et al., 2019) and PyTorch Geometric framework 1.7.2 (Fey & Lenssen, 2019). The details of each dataset is shown in Table. 10, all of the real datasets are publicly available, and we follow the code in (Toenshoff et al., 2021) to generate the RB model. The real-world dataset split follows that in Karalias & Loukas (2020).

To balance the training time per epoch of EGN and Meta-EGN, we define the epoch as follows: For each epoch of EGN training, the whole dataset is split into mini-batches. EGN performs standard

Table 10: The number of instances in each dataset. '20/scale/degree' means that we generate 20 testing instances for each different scale-degree pair. We generate RB1000, RB2000, and RB5000 only for testing.

| Dataset | Twitter | COLLAB | IMDB | RB200 | RB500 | RB1000 | RB2000 | RB5000 | RRGs |
|---|---|---|---|---|---|---|---|---|---|
| Training | 665 | 3150 | 700 | 2000 | 2000 | - | - | - | 3000 |
| Validation | 95 | 450 | 100 | 100 | 100 | - | - | - | 15 |
| Testing | 190 | 900 | 200 | 100 | 100 | 100 | 100 | 100 | 20/scale degree |

mini-batch training along these batches and optimizes over each mini-batch. As to Meta-EGN, for each training epoch Meta-EGN only randomly samples a single batch and does the meta learning algorithm on the batch. The batch sizes of the methods are controlled the same.

## C.2 DETAILED DERIVATION OF THE LOSS FUNCTION RELAXATION

In this part, we display the detailed loss function relaxation of the three problems in our study (the MC, the MVC, and the MIS). The basic idea of training loss design and relaxation follow (Karalias & Loukas, 2020; Wang et al., 2022). In the following derivation, we use $i, j$ to represent the nodes in graphs, we use $X_i, X_j \in \{0, 1\}$ to denote the discrete assignment of the binary optimization variables, and we use $\bar{X}_i, \bar{X}_j \in [0, 1]$ to denote the relaxed soft assignment of the binary optimization variables.

**The maximum clique (MC):** A clique is a set of nodes $S \in V$ such that any two distinct nodes in the set are adjacent. The MC aims to find out the clique with the largest number of nodes. We could formulated the optimization objective as follows:

$$\max_X \sum_{1 \le i \le n} X_i \qquad \text{s.t.} \quad (i, j) \in E \text{ if } X_i, X_j = 1, \tag{16}$$

$X_i, X_j$ denotes whether to take the node into the clique set ($X_i = 1$) or not ($X_i = 0$). By setting a proper penalty coefficient $\beta$, we could formulate the loss function relaxation as follows (the detailed derivation follows the corresponding case study in Karalias & Loukas (2020)).

$$l_{\text{MC}}(\theta; G) \triangleq -(\beta + 1) \sum_{(i,j) \in E} \bar{X}_i \bar{X}_j + \frac{\beta}{2} \sum_{i \ne j} \bar{X}_i \bar{X}_j. \tag{17}$$

**The minimum vertex covering (MVC):** A vertex cover is a set of nodes $S \in V$ that any edge in the graph is connected to at least a node from the set. The MVC aims to find out the cover set with the smallest number of nodes. The optimization objective could be summarized as follows:

$$\min_X \sum_{1 \le i \le n} X_i \qquad \text{s.t.} \quad X_i + X_j \ge 1 \text{ if } (i, j) \in E, \tag{18}$$

where $X_i, X_i$ denotes whether to take the node into the cover set ($X_i = 1$) or not ($X_i = 0$). We design the constraint function $g$ to represent the total number of edges that have not been covered given a set of variable assignment $X$, and thus we write $g$ as:

$$g_{\text{MVC}}(X; G) \triangleq \sum_{(i,j) \in E} (1 - X_i)(1 - X_j). \tag{19}$$

Then we relax the constraint $g$ and add it into the training objective by multiplying a proper penalty coefficient $\beta$, following the relaxation principle in Wang et al. (2022):

$$l_{\text{MVC}}(\theta; G) \triangleq \sum_{1 \le i \le n} \bar{X}_i + \beta \sum_{(i,j) \in E} (1 - \bar{X}_i)(1 - \bar{X}_j). \tag{20}$$

By this, we aim to minimize the value of the loss function above in order to minimize the node number of the cover set as well as consider the covering property in the constraint.

**The maximum independent set (MIS):** An independent set is a set of nodes where any two distinct nodes in the set are not adjacent to each other. The MIS aims to find out the independent set with the largest number of nodes. We could formulate the objective of the MIS as follows:

$$\max_X \sum_{1 \le i \le n} X_i \qquad \text{s.t.} \quad X_i X_j = 0 \text{ if } (i, j) \in E, \tag{21}$$

where $X_i, X_j$ denotes whether to take the node into the independent set ($X_i = 1$) or not ($X_i = 0$). We formulate the constraint $g$ as the total number of edges whose two connected nodes at the end points are both assigned into the independent set. Therefore we could write the constraint as follows:

$$g_{\text{MIS}} \triangleq \sum_{(i,j) \in E} X_i X_j. \tag{22}$$

We then relax the constraint $g$ into continuous space and add it into the c function with a proper penalty coefficient $\beta$, following the relaxation principle in (Wang et al., 2022), and thus we could write the training loss function as:

$$l_{\text{MIS}}(\theta; G) \triangleq - \sum_{1 \leq i \leq n} \bar{X}_i + \beta \sum_{(i,j) \in E} \bar{X}_i \bar{X}_j. \tag{23}$$

By this, we aim to minimize the value of the loss function above in order to maximize the node number of the independent set as well as consider the independent property in the constraint.

### C.3 SEPARATED ALGORITHM TABLES

We separate the algorithm table of Meta-EGN into training and testing parts to make it clearer. The algorithm table is shown in Alg. 2 for training and Alg. 3.

---

**Algorithm 2** Train Meta-EGN

---

**Require:** Training instances $\Xi = \{G_1, G_2, ..., G_m\}$; Hyperparameters: $\alpha, \gamma$.
 1: Randomly initialize $\theta^{(0)}$
 2: **for** each randomly sampled mini-batch $B_j \subset \Xi, j = 0, 1, ..., K-1$ **do**       ▷ Training starts
 3:     For each $G_i \in B_j$, compute the adapted parameter: $\theta_i^{(j)} = \theta^{(j)} - \alpha \nabla_{\theta^{(j)}} l(\theta^{(j)}; G_i)$
 4:     Update: $\theta^{(j+1)} \leftarrow \theta^{(j)} - \gamma \nabla_{\theta^{(j)}} \sum_{G_i \in B_j} l(\theta_i^{(j)}; G_i)$
 5: **end for**
 6: **return** $\theta \leftarrow \theta^{(K)}$       ▷ Training ends

---

---

**Algorithm 3** Test Meta-EGN with/without Fine-tuning

---

**Require:** Testing instance $G'$; Hyperparameter: $\alpha$; Pre-trained parameter initialization $\theta$.
 1: For a given testing instance $G'$:       ▷ Testing starts
 2: **if** fine-tuning is allowed **then**
 3:     Fine-tune the parameters: $\theta_{G'} \leftarrow \theta - \alpha \nabla_\theta l(\theta; G')$
 4:     Use Def. 1 to round the relaxed solution given by $\mathcal{A}_{\theta_{G'}}(G')$       ▷ With fine-tuning
 5: **else**
 6:     Use Def. 1 to round the relaxed solution given by $\mathcal{A}_\theta(G')$       ▷ Without fine-tuning
 7: **end if**
 8: **return** the rounded solution       ▷ Testing ends

---

### C.4 IMPLEMENTATION OF THE HEURISTICS

We run all of the greedy algorithms with PyThon 3.8 in this paper. A potential method to boost the time cost of these greedy algorithms is to use c++.

**Random Greedy Algorithm for MIS (RGA):** RGA takes a time to reach a solution that is linear in the problem size n. It starts from an empty independent set $S$. At each step $1 \leq t \leq n$, a node $i$ is chosen at random from the graph $G_t$ and added to the independent set. Then all the neighbors of $i$ are removed from $G_t$ to formulate a new graph $G_{t+1}$. The process iterates until $G_{t^*}$ is empty at step $t^*$, the solution is $S$.

**Degree-based Greedy Alforithm for MIS (DGA):** DGA modifies RGA by sorting the degrees of the nodes before each iteration starts, and always put the node with the smallest degree into the independent set.

**Teonshoff Greedy for MC:** Toenshoff et al. (2021) convert the testing instances into its complement graph, and then run DGA to solve the MIS problem. It takes the solution to the MIS problem on the complement graph as the solution for MC on the original graph.

**Greedy for MVC:** Greedy for MVC starts from an empty covering set $S$. At each step $1 \le t \le n$, it first sorts the degrees of the nodes in the graph $G_t$ and always adds the node $i$ with the largest degree into the covering set $S$. Then all the edges that connect with $i$ are removed from $G_t$ to formulate a new graph $G_{t+1}$. The process stops until $G_{t^*}$ is empty at step $t^*$, the solution is $S$.

## D    DISCUSSION ON LIMITATIONS

• As mentioned at the end of Sec. 5.1, both EGN and Meta-EGN perform generally well in MC, which outputs the cliques that are more local in comparison with the vertex covering in MVCs that require more global assignments. The random initialization seed with one node randomly set as $1$ and the others as $0$ would potentially limit the performance of EGN and Meta-EGN in more global CO tasks.

• We use Meta-EGN and EGN to modify the solution of DGA and RGA in the MIS problem. In addition, there are also many other Monte Carlo (MC) algorithms (i.e. simulated annealing and parallel tempering) that could produce better results than DGA or RGA in RRGs (Angelini & Ricci-Tersenghi, 2022). An intuitive idea is to test whether we could learn Meta-EGN to further fine-tune these more advanced MC algorithms in the MIS problem on RRGs.

We leave the research on modifying Meta-EGN to better deal with the CO problems that require global assignments and using Meta-EGN to improve other advanced MC algorithms as a future study.

