# OpenReview forum: "Unsupervised Learning for Combinatorial Optimization Needs Meta Learning"
_ICLR.cc/2023/Conference — ICLR 2023 poster_

### Official Review · Reviewer_uyeg · 2022-10-21

**Confidence:** 4
**Correctness:** 4
**Technical Novelty And Significance:** 2
**Empirical Novelty And Significance:** 3
**Recommendation:** 6

**Clarity, Quality, Novelty And Reproducibility:**

Clarity: The paper is generally straight forward and clearly written. I have several questions below:

-- How does one obtain the functions $f_r$ and $g_r$ given any arbitrary $f$?

-- Are the relaxations in Table 2 well-known? If this has been discussed in previous work, the authors should cite give a brief summary. Otherwise, please provide some intuitions regarding their constructions.

-- How is pareto-optimality defined in this paper?

-- Were EGN and Meta-EGN pretrained for the same amount of time? This is an important detail that should be in the description, but I can't seem to find any mention. One update iteration of meta learning is a generally more costly and samples a lot more data than one standard update iteration. The only way to fairly compare a model and a meta-model is to make sure that they were trained for the same amount of time and have seen roughly the same amount of samples.

-- The section title "META-EGN BOOSTS THE PERFORMANCE WITHOUT DISTRIBUTION SHIFTS" seems a bit misleading since Gurobi actually performs optimally on more than half the benchmarks. I guess the point is that Gurobi takes 4s per task, whereas it only takes <1s  for the pretrained models to do inference. This difference will eventually accumulate in repeated inference scenarios. If this is the case, the author should also discuss the break-even point where the pretrained models are per-task cheaper than Gurobi (which also means reporting the pretraining time).

Novelty: This is a direct application of MAML on EGN. While the exact problem has not been solved before, the idea is not surprising since it is subsumed by the model agnostic design of MAML.

Quality: The result for smaller graphs seems marginal. The result for generalization on larger graph is good. For the MIS case I think building on top of the DGA result is a valid strategy. However, to be fair, the DGA algorithm should be allowed to continue running from that point for the same amount of time allocated to Meta-EGN (I'm under the impression that the DGA algorithm will continue to run until the exact solution is found -- please correct me if i am wrong). This has not been discussed in the description.

Reproducibility: I'm convinced that this paper can be reproduced. The result seems to align with general trend in meta learning and previous result of EGN.

**Strength And Weaknesses:**

Strengths: Interesting application with a straight-forward solution. The authors provide a theoretical guarantee and obtain significant result on several classical problems.

Weaknesses:

-- Some missing important details, please check the questions below.

-- Running time, including the pretraining time, should be thoroughly reported for fair comparison.

-- It is unclear what should be the take away insight of this paper. EGN has been shown to be a robust CO solver that can generalize. Meta learning has been shown to improve generalization in many applications. While the results in this paper are mostly positive, I feel that they only serve to confirm previous insights.

**Summary Of The Paper:**

This paper proposes an application of Model-agnostic meta learning (MAML) (Finn et al., 2017) for the Erdos-Goes-Neural (EGN) framework (Karalias and Loukas, 2020). This will allow faster learning of novel combinatorial optimization tasks through learning better model initialization. The method is applied to several classical problems such as Max clique (MC), Minimum vertex covering (MVC) and Max independent set (MIS). The empirical result is promising.

**Summary Of The Review:**

This paper presents an interesting application of meta-learning on the EGN framework. There are some positive results, although they are not quite surprising. I have no major concerns about the quality of the paper. Overall, I would recommend a marginal acceptance.

---

> ### Author Response · Authors · 2022-11-14
> **Response to reviewer uyeg (1/3)**
>
> Thank you so much for reviewing our paper, appreciating our solutions and raising the valuable comments. Here we address the questions in the following response.
>
> > Running time, including the pretraining time, should be thoroughly reported for fair comparison.
>
> Thanks for your suggestion. We report the training time from start to convergence (the model achieves the best performance on the validation set) in Table.9 in the Appendix and in the following table. The training wall clock time to convergence of Meta-EGN is basically longer than EGN by 1-2 times. But the total training time that they require to converge are basically on the same order of magnitude.
>
> | Dataset/Time (min:second) |    MC   |        |        |   MVC   |        |        |   MIS   |
> |:-------------------------:|:-------:|:------:|:------:|:-------:|:------:|:------:|:-------:|
> |                           | Twitter |  RB200 |  RB500 | Twitter |  RB200 |  RB500 |   RRGs  |
> |            EGN            |  46:50  | 104:37 | 282:57 |  100:58 |  83:27 | 128:39 |  733:02 |
> |          Meta-EGN         |  101:55 | 210:04 | 609:47 |  276:38 | 168:25 | 282:15 | 1088:55 |
>
> But here we would like to point out that in real-world applications, the aim of learning for CO is mainly to have fast inference[1][3][4][5]. Because once the training finishes, the model could be used to do inference repetitively. When the application asks to solve the problem repetitively, classical CO solvers are not good choices because they may not leverage historical data and generally take much more inference time compared with learning-based methods. Also note that all previous RL works often take tens or hundreds of more time than the training method adopted in this work, which means if training time was counted, all these RL methods take a so long time and lose their competitiveness.
> Thus we do not think we should place the training time into a prominent place when we compare the performance of learning methods for CO.
>
> > It is unclear what should be the take away insight of this paper. EGN has been shown to be a robust CO solver that can generalize. Meta learning has been shown to improve generalization in many applications. While the results in this paper are mostly positive, I feel that they only serve to confirm previous insights.
>
> Our key takeaway insights could be summarized in the following three folds.
>
> - From the perspective of problem objective formulation, we are the first to propose  learning with per-instance optimal objective instead of learning with averaged performance. Because of this, we are also the first one to formulate the unsupervised learning for CO problems with the meta-learning framework. Such knowledge may further guide the objective design for the other unsupervised learning methods.
>
> - From the perspective of model generalization, Meta-EGN is the first to apply meta-learning to handle CO objectives. Moreover, we observe interesting behaviors  that even if Meta-EGN does not do fine-tuning during the testing, Meta-EGN can still outperform EGN. This can be hardly seen in other meta-learning scenarios.
>
> - From the node feature input and the expressive power aspect. We are the first to consider utilizing the output of heuristic methods as the node feature initialization for GNNs in the MIS. To the best of our knowledge, our idea of using neural networks to learn how to fine-tune the heuristic outputs is novel and has further research potential, it may further inspire the research in the relationship between the graph node feature initialization and the expressive power of neural networks in learning for CO.

---

> > ### Author Response · Authors · 2022-11-14
> > **Response to reviewer uyeg (2/3)**
> >
> > > How does one obtain the functions fr and gr given any arbitrary f?
> >
> > > Are the relaxations in Table 2 well-known? If this has been discussed in previous work, the authors should cite give a brief summary. Otherwise, please provide some intuitions regarding their constructions.
> >
> > We answer the two questions above together in the following response. The training loss relaxation basically follows the case study in [1] and the relaxation principle in [2].
> > For any given binary CO problem, one could always write its corresponding cost function $f(X;G)$ and constraint $g(X;G)$ into an entry-wise affine form (See the case studies in [1] and Theorem. 2 in [2]), where $X\in \{0,1\}^n$. Then one could directly relax the functions $f(X;G)$ and $g(X;G)$ from discrete space $X\in \{0,1\}^n$ into the continuous space as $f_r(\bar{X};G)$ and $g_r(\bar{X};G)$, where $\bar{X} \in [0,1]^n$ (see Equation.3 in [1] and Equation.3 in [2]).
> > With the relaxed $f_r$ and $g_r$, one could choose a penalty coefficient $\beta > \max f(X;G)$ and construct the loss function $l_r = f_r(X;G) + \beta g_r(X;G)$. It’s proved that training with such a relaxed loss function could achieve performance guarantee as long as the loss value is small enough on an encountered testing instance (See Theorem. 1 in [1] and Theorem. 1 in [2]).
> > We did not give a clear summary due to the space limit, now we cite the relative papers in the revised manuscript and added the derivation in Appendix C.2.
> >
> > > How is pareto-optimality defined in this paper?
> >
> > Sorry that we did not make this point clear. The bolded texts represent pareto-optimal solutions. The pareto-optimal solution denotes the solutions that: within the time budget, the method achieves the best performance; Or to achieve such performance, the method requires the least time. In other words, other methods must require more time to further improve the performance, or other methods would not provide a better solution with less time. We refined the bolded text in Table.3, 4 to make it clearer.
> >
> > > One update iteration of meta learning is a generally more costly and samples a lot more data than one standard update iteration. The only way to fairly compare a model and a meta-model is to make sure that they were trained for the same amount of time and have seen roughly the same amount of samples.
> >
> > Meta-EGN is indeed more costly per iteration. As shown in the table above, the training wall clock time to convergence of Meta-EGN is basically longer than EGN by 1-2 times. But the total training time that they require to converge are basically on the same order of magnitude. Moreover, as mentioned above, in learning for CO community, one may pay more attention to the testing performance (time per testing instance together with the approximation rate) than the training time. Because after the one-time training, the model could be used repetitively in any encountered testing instance in the future, and the application scenario for learning for CO generally requires relatively short time which might not be enough for the classical solver cold start.
> > In addition, to mention that compared with previous reinforcement learning (RL), both meta-EGN and EGN require much less time to converge.
> >
> > > If this is the case, the author should also discuss the break-even point where the pretrained models are per-task cheaper than Gurobi (which also means reporting the pretraining time).
> >
> > When we only consider the required testing time for learning-based methods and classical solvers, as the distribution becomes more complicated or the graph size becomes larger, it generally costs more inference time for classical solvers than learning-based methods. Thus such break-points depend on both the distribution complexity and the graph size.
> >
> > In addition, we do not think we should take the training time into consideration when we compare against the classical solvers. The reason is that the application scenarios for learning based methods generally use the models to repetitively solve testing instances and ask for no training time limitation, but instead require relatively good solutions within a short inference time budget.

---

> > > ### Author Response · Authors · 2022-11-14
> > > **Response to reviewer uyeg (3/3)**
> > >
> > > > However, to be fair, the DGA algorithm should be allowed to continue running from that point for the same amount of time allocated to Meta-EGN (I'm under the impression that the DGA algorithm will continue to run until the exact solution is found -- please correct me if i am wrong).
> > >
> > > DGA algorithm is the greedy algorithm that selects nodes in sequence according to the node degree. Thus the running time of DGA is finite and fixed.
> > >
> > > But as the reviewer points out, we may consider fine-tuning the result of other methods (e.g. the monte carlo methods that won’t stop till an optimal solution is found), we leave this as a future study.
> > >
> > > [1] Karalias et al. Erdos goes neural: an unsupervised learning framework for combinatorial optimization on graphs. NeurIPS 2020.
> > >
> > > [2] Wang et al. Unsupervised Learning for Combinatorial Optimization with Principled Objective Relaxation. NeurIPS 2022.
> > >
> > > [3] Silver et al. Mastering the game of Go without human knowledge. Nature 2017.
> > >
> > > [4]Minh et al. Human-level control through deep reinforcement learning. Nature 2015.
> > >
> > > [5]Khali et al. Learning combinatorial optimization algorithms over graphs. NeurIPS 2017.

---

> > > > ### Comment · Reviewer_uyeg · 2022-11-24
> > > > **My score will remain positive**
> > > >
> > > > I thank the authors for taking the time to answer my questions. I'm convinced by the new results and my score will remain positive.

---

> > > > > ### Author Response · Authors · 2022-11-25
> > > > > **Further response to reviewer uyeg**
> > > > >
> > > > > Dear reviewer uyeg,
> > > > >
> > > > > Thank you again for your time to check the results and remain positive in the further response.
> > > > >
> > > > > Best,
> > > > >
> > > > > authors.

---

### Official Review · Reviewer_sUXy · 2022-10-24

**Confidence:** 3
**Correctness:** 3
**Technical Novelty And Significance:** 3
**Empirical Novelty And Significance:** 3
**Recommendation:** 8

**Clarity, Quality, Novelty And Reproducibility:**

There are a few typos but the paper is clear to follow and adequate information is available for reproducibility.

**Strength And Weaknesses:**

As demonstrated in the experiments section, Meta-EGN works well compared to all relevant baselines, except for Gurobi of course, which is known to be in a league of its own compared to any neural CO solver. It is fascinating that such a general parameter $\theta$ can even exist for these problems.

The ability to extend the performance guarantees of EGN toMeta-EGN is also a good win.

One drawback is missing ablation studies about the technique - some questions that I would be interested in are

- How does Meta-EGN perform as a function of %of the training data used? i.e., if we initialize algorithm 1 with random subsets of the full training dataset of size N/2 instead N and training Meta-EGN to convergence, how does the algorithm perform?
- How does Meta-EGN generalize if it is trained on graphs with certain statistics i.e. average node degree < 3 and then evaluated on graphs with avg node degree > 3? Or similar studies.

Ablation studies like these would help answer how sensitive Meta-EGN is to the quality of the training data set, my hypothesis is that if the training data set is not diverse enough, Meta-EGN wont work as well.

>  the current framework optimizes an averaged performance over the distribution of historical problem instances, which misaligns with the actual goal of CO that looks for a good solution to every future encountered instance'

A note on this comment - in practice - we don't wont want to solve the general case problem cause often CO problems can be intractable (eg NP Hard). Instead, practitioners would develop a heuristic to try to exploit a property that is only true in the distribution of instances that they are interested. eg. TSP is hard in general but approx algos exist in a metric space -- It is unclear from the paper what a practitioner should do in this case - should they use a training dataset that matches the distribution of instances that they are interested in OR should they use a more general distribution of instances since it will allow Meta-EGN to learn general case properties about the problem which wont be visible in the specific distribution.

An excellent version of the paper would offer more direction to practitioners on how to use the framework, however this could be tackled in future work.


**Summary Of The Paper:**

This paper presents a meta learning framework Meta-EGN that produces a good initialization for a neural CO solver. The authors build upon the work of Karalias & Loukas, (2020) and extend it to provide a performance guarantee for Meta-EGN as well.

**Summary Of The Review:**

Overall I found the paper interesting and it poses some fun questions for future work. The novelty is a little incremental since it builds on top of recent work but the results are interesting.

---

> ### Author Response · Authors · 2022-11-14
> **Response to reviewer sUXy (1/2)**
>
> Thank you so much for reviewing our paper, appreciating our idea, and raising the actionable suggestions as well as the insightful questions. Here we address the questions in the following response.
> > How does Meta-EGN perform as a function of %of the training data used? i.e., if we initialize algorithm 1 with random subsets of the full training dataset of size N/2 instead N and training Meta-EGN to convergence, how does the algorithm perform?
>
> That is a good suggestion to observe the sensitivity of Meta-EGN. We further conduct experiments on the max clique problem on the twitter dataset. The experimental results are shown in the following table and in Table.8 in the Appendix of the revised manuscript.
>
> | training size |      64     |     128     |     256     |     512     |   Full 750  |
> |:-------------:|:-----------:|:-----------:|:-----------:|:-----------:|:-----------:|
> |      EGN      | 0.909±0.122 | 0.911±0.118 | 0.914±0.118 | 0.922±0.115 | 0.926±0.113 |
> |    Meta-EGN   | 0.970±0.058 | 0.973±0.055 | 0.975±0.055 | 0.975±0.051 | 0.976±0.048 |
>
> As shown in the table above, both methods would suffer some performance decay when trained with fewer training instances. However, Meta-EGN appears to be more robust against a fewer number of training samples compared with EGN.
>
> > How does Meta-EGN generalize if it is trained on graphs with certain statistics i.e. average node degree < 3 and then evaluated on graphs with avg node degree > 3? Or similar studies.
>
> That’s also a good suggestion to further investigate the generalization of Meta-EGN. We train EGN and Meta-EGN only on RRGs with node degree 3 or RRGs with node degree 20 and test the models on the testing instances with node degrees of 3,7,10,20. We train the models with the output of DGA as input graph features. We display the results without fine-tuning in Fig. 7 in Appendix in the revised manuscript. We show the table version below.
>
> Models are trained on RRGs with degree 3 only:
> | trained on degree 3 | num nodes/ degree |    3   |    7   |   10   |   20   |
> |:-------------------:|:-----------------:|:------:|:------:|:------:|:------:|
> |         EGN         |        10^3       | 0.9525 | 0.7712 | 0.6553 | 0.6238 |
> |                     |        10^4       | 0.9524 | 0.7634 | 0.6602 | 0.6309 |
> |       Meta-EGN      |        10^3       | 0.9529 | 0.8703 | 0.7687 | 0.6248 |
> |                     |        10^4       | 0.9526 | 0.8818 | 0.7993 | 0.6319 |
>
> Models are trained on RRGs with degree 20 only:
> | trained on degree 20 | num nodes/ degree |    3   |    7   |   10   |   20   |
> |:--------------------:|:-----------------:|:------:|:------:|:------:|:------:|
> |          EGN         |        10^3       | 0.9525 | 0.9046 | 0.9009 | 0.8716 |
> |                      |        10^4       | 0.9529 | 0.9107 | 0.9013 | 0.8796 |
> |       Meta-EGN       |        10^3       | 0.9535 | 0.9093 | 0.9026 | 0.8726 |
> |                      |        10^4       | 0.9540 | 0.9146 | 0.9027 | 0.8797 |
>
>
> It is shown that when only trained on RRGs with degree 3, both Meta-EGN and EGN could not generalize well, while Meta-EGN still outperforms EGN in all the testing cases. The phenomenon aligns with the reviewer’s conjecture to some extent that if Meta-EGN has never met instances ever similar with the testing data during the training process, it may not work very well. However, Meta-EGN still improves EGN regarding the model generalization.
> Surprisingly, when we train the models only on RRGs with degree 20, both the methods tend to achieve relatively good generalization in the testing set with 3,7,10 degrees. We attribute this result to the fact that RRGs with degree 20 are much more complicated than those with only degree 3, and neural networks may learn heuristics from RRGs with degree 20 and generalize to RRGs with fewer degrees. Note that in this case, Meta-EGN also outperforms EGN.

---

> > ### Author Response · Authors · 2022-11-14
> > **Response to reviewer sUXy (2/2)**
> >
> > > It is unclear from the paper what a practitioner should do in this case - should they use a training dataset that matches the distribution of instances that they are interested in OR should they use a more general distribution of instances since it will allow Meta-EGN to learn general case properties about the problem which wont be visible in the specific distribution.
> >
> > We sincerely appreciate this insightful question. We think this belongs to an open-ended question that is valuable to further research and discussion. Here, we try to explain it based on our current understanding.
> > We analyze the question by splitting it into two cases.
> >
> > In the first case where people know about the testing distribution ahead of training, they might construct the training dataset with the same distribution as the testing distribution. Meta-EGN might achieve better per-instance performance due to its training objective for per-instance solutions. Our experiments in the MIS on RRGs with degrees 3,7,10,20 could further stand for this point of view: in the case when we know that the testing instances consist of RRGs with degrees 3,7,10,20, we construct the training dataset with the corresponding distribution and Meta-EGN could outperform EGN.
> >
> > In the second case where people do not know about the testing distribution, we suggest that they might take the training dataset with a diverse distribution given the same training sample budget. This may increase the probability for Meta-EGN to learn some heuristics from the historical data and thus generalize better to the testing data.
> >
> > But we have the same conjecture that if the testing distribution is totally different from the training distribution, even Meta-EGN could not generalize well, as it has never seen similar instances. This argument could also be supported by the supplementary experiments when the models are only trained on RRGs with degree 3.

---

> > > ### Comment · Reviewer_sUXy · 2022-11-26
> > > **Response**
> > >
> > > Thank you for responding to my queries!
> > >
> > > I retain my recommendation to accept this paper

---

> > > > ### Author Response · Authors · 2022-11-27
> > > > **Thanks for checking our response**
> > > >
> > > > Dear reviewer sUXy,
> > > >
> > > > We appreciate your time for checking our results and response. Thank you for remaining positive in the further response.
> > > >
> > > > Best,
> > > >
> > > > authors

---

### Official Review · Reviewer_oMcq · 2022-10-25

**Confidence:** 4
**Correctness:** 2
**Technical Novelty And Significance:** 3
**Empirical Novelty And Significance:** 2
**Recommendation:** 8

**Clarity, Quality, Novelty And Reproducibility:**

## Questions

According to the proof of Theorem 1, $\alpha$ is assumed as $1/L$. I think that it should be stated more clearly. Also, a learning rate should be determined as $1/L$. Did you consider this value when you choose a learning rate?

Following the above question, Theorem 1 can be proved where a stochastic gradient descent is used. Could I ask which optimizer is used?

I want to ask about bold texts in the tables in this work. I think bold texts are somewhat randomly determined. Does a bold text represent the best result?

In Table 1, what is the meaning of fine-tuning timing for classical solver? Sine it does not have a training procedure, fine-tuning does not also exist, right?

## Minor issues

In Page 3, "a algorithm" should be "an algorithm" and "a NN" should be "an NN".

I think that Algorithm 1 should be separated to "train" and "test". "return" in Line 6, which is in the middle of the algorithm is very strange to me.

**Strength And Weaknesses:**

## Strengths

This paper is generally well-written, and the main message of this work is clear. Moreover, the proposed algorithm shows good performance in many benchmarks.

## Weaknesses

The novelty of this work is somewhat limited. However, I think that this does not degrade the quality of the paper much.

Moreover, I think the sentence

> our learning objective of a model is to make further optimization of the initialization given by this model over each of these pseudo-new instances yield good solutions

should be proved. I think that the reasoning for this statement is not provided appropriately.

Also, this work heavily relies on the recent work (Wang et al., 2022). Some parts of this work are very similar with the work (Wang et al., 2022).

**Summary Of The Paper:**

This work suggests a framework of unsupervised learning for combinatorial optimization, which uses generally good initializations that determine via a meta-learning scheme. In particular, the authors propose a new objective of unsupervised learning that is capable of providing good initializations rather than direct solutions. Finally, the authors demonstrate the empirical results that show the effectiveness of the proposed method.

**Summary Of The Review:**

Please see the above text boxes.

---

> ### Author Response · Authors · 2022-11-14
> **Response to reviewer oMcq**
>
> Thank you so much for reviewing our paper, appreciating the quality of our paper and providing the detailed and actionable suggestions. Here we address the confusions and questions in the following response.
>
> >‘*our learning objective of a model is to make further optimization of the initialization given by this model over each of these pseudo-new instances yield good solutions*’ should be proved. I think that the reasoning for this statement is not provided appropriately.
>
> Sorry for the confusion. We have rephrased this sentence in the revised manuscript as:
> *our learning objective is to learn a good initialization of this model, such that further optimization of the initialization could achieve good solutions on each of these pseudo-new instances.*
> This sentence formulates our insight and objective of the learning for CO problem, such objective naturally aligns with that of meta-learning. And thus we adopt the MAML technique into our framework. The sentence itself is not a theoretical argument, it is the motivation of our objective formulation. We are not sure about which type of proof **Reviewer oMcq** may expect.
>
> > this work heavily relies on the recent work (Wang et al., 2022). Some parts of this work are very similar with the work (Wang et al., 2022).
>
> We respectfully disagree with the statement. Indeed we use the notations from [1] and make similar assumptions as the recent work[1]. But the two works essentially focus on different problems.
>
> [1] studies how to relax the CO objectives that allow some performance guarantee. And it extends its relaxation-and-rounding principle from pure CO problems to procyCO problems.
>
> However, motivated by the gap between the average performance and the actual goal for a good solution on every encountered testing instance, we study how such per-instance optimality as the objective can further improve the performance of learning for CO.
>
> > Did you consider this value when you choose a learning rate?
>
> The short answer is No.
>
> Our theorem is to extend the performance guarantee to the setting of our proposed objective. However, in practice, we choose a manually tuned inner learning rate. We guess the Lipschitz constant is hard and costly to evaluate in practice, though it might be useful to guide the tuning of the inner learning rate. We leave the empirical study of this in the future.
>
> > Theorem 1 can be proved where a stochastic gradient descent is used. Could I ask which optimizer is used?
>
> We follow the proof in Theorem 1 in our implementation. In our meta-EGN framework, we view each training instance as an independent task and use gradient descent on each instance in the inner optimization loop.
>
> > I think bold texts are somewhat randomly determined. Does a bold text represent the best result?
>
> The bolded texts represent pareto-optimal solutions. The pareto-optimal solution denotes the solutions that: within the time budget, the method achieves the best performance; Or to achieve such performance, the method requires the least time. In other words, other methods would not provide a better solution with less time. We refine the bolded text in Table.3, 4 to make it clearer.
>
> > what is the meaning of fine-tuning timing for classical solver? Since it does not have a training procedure, fine-tuning does not also exist, right?
>
> As the classical solver does not need training, in order to align with the comparison among the other learning based methods, we use the phrase `fine-tuning’ time to denote the time that the classical solver requires to give the solution for each testing instance.
>
> > I think that Algorithm 1 should be separated to "train" and "test". "return" in Line 6, which is in the middle of the algorithm is very strange to me.
>
> Sorry for the confusion. We combined the training and testing procedure together due to the space limit of 9 pages. We now refine the original Algorithm table and separate the Algorithm into ‘training’ and ‘testing’. Please refer to appendix C.3 in the revised version.
>
> [1] Wang et al. Unsupervised Learning for Combinatorial Optimization with Principled Objective Relaxation. NeurIPS 2022.

---

> > ### Comment · Reviewer_oMcq · 2022-11-21
> > **Thank you for your response**
> >
> > Hi Authors,
> >
> > Thank you for your response.
> >
> > I think that some concern still remains.  In particular, I have a concern on the setting of learning rate.  If the configuration of learning rate did not follow the theoretical results, I think a thorough numerical analysis should be provided.

---

> > > ### Author Response · Authors · 2022-11-25
> > > **Further response to reviewer oMcq**
> > >
> > > Dear reviewer oMcq,
> > >
> > > Thanks so much for your further actionable suggestions. We would like to address the concern from both the theoretical side and the empirical side:
> > >
> > > $\bullet$ From the theoretical aspect, although Theorem 1 in the paper states that $\alpha$ is set as $1/L$ to achieve the performance guarantee for Meta-EGN,  it is not the only value that may work. Here, we would like to extend Theorem 1 to a more general condition of $\alpha$. Specifically, we could prove that for any $\alpha < 2/L$, the performance guarantee for Meta-GNN still exists. The rigorous theorem is as follows:
> > >
> > > Suppose the relaxations $f_r$ and $g_r$ are entry-wise concave. Let $\theta$ denote the learned parameter after training.  Given a test instance $G$, suppose locally $l(\cdot;G)$ is $L$-smooth at $\theta$, i.e., $\|\|\nabla_{\theta'} l(\theta';G) - \nabla_{\theta} l(\theta;G)\|\|\leq L\|\|\theta' - \theta\|\|$ for all $\theta'$ that satisfies $\|\|\theta' - \theta\|\|\leq \epsilon$. Then, if $l(\theta;G) < \beta + \triangle$ (even if $l(\theta;G) \geq \beta$), for any $\alpha \in (0, 2/L)$ Meta-GNN with one-step finetuning outputs a feasible solution $X$ of good quality $f(X;G)\leq l(\theta;G)-\triangle$. Here, $\triangle = \|\|\nabla_{\theta} l(\theta;G)\|\| \epsilon + \frac{1}{2L\alpha^2 - 4\alpha}\epsilon^2$ if $\epsilon < \alpha \|\|\nabla_{\theta} l(\theta;G)\|\|$ or $\triangle = (\alpha - \frac{L\alpha^2}{2})\|\|\nabla_{\theta}l(\theta;G)\|\|^2$ o.w..
> > >
> > > The proof is similar to that of Theorem 1. The only difference is that here we no longer consider the best $\alpha$ value selection, instead we consider a much wider range for more general cases. **Such extension of the theorem could provide more flexibility to the choice of $\alpha$ in theory.** We would like to enclose this more general theorem in our final revised version.
> > >
> > > $\bullet$ As to the empirical aspect, we run Meta-EGN with the inner learning rate perturbed around the original learning rate to show the robustness of our framework with respect to $\alpha$. We do experiments on Twitter for the max clique, RB200 for the minimum vertex covering and RRGs with $10^4$ nodes for the max independent set. The results are shown as follows:
> > >
> > > |    ApR / Inner lr   |      3e-5      |      5e-5      |      7e-5      |      1e-4      |
> > > |:-------------------:|:-------------:|:-------------:|:-------------:|:-------------:|
> > > |    MC on Twitter    |  0.980±0.0519 |  0.976±0.0483 |  0.975±0.0494 |  0.974±0.0552 |
> > > |     MVC on RB200    |  1.028±0.0057 |  1.028±0.0055 |  1.028±0.0057 |  1.029±0.0060 |
> > > |  MIS on RRG d=3 | 0.9536±0.0019 | 0.9544±0.0017 | 0.9535±0.0018 | 0.9512±0.0002 |
> > > |  MIS on RRG d=7 | 0.9189±0.0023 | 0.9191±0.0022 | 0.9189±0.0023 | 0.9188±0.0022 |
> > > | MIS on RRG d=10 | 0.9088±0.0029 | 0.9088±0.0029 | 0.9088±0.0030 | 0.9085±0.0030 |
> > > | MIS on RRG d=20 | 0.8859±0.0040 | 0.8862±0.0038 | 0.8858±0.0036 | 0.8854±0.0041 |
> > >
> > > It’s shown that empirically the performance of Meta-EGN is robust with respect to the choice of $\alpha$ to some extent.

---

> > > > ### Comment · Reviewer_oMcq · 2022-11-25
> > > > **Thanks for your follow-up response**
> > > >
> > > > Thank you for your response.
> > > >
> > > > The rigorous theorem looks interesting and I think it will help to understand your paper better.
> > > >
> > > > Also, the empirical results are interesting.  The results seem quite robust across learning rates.
> > > >
> > > > Since my concern has been resolved, I will increase my score.

---

> > > > > ### Author Response · Authors · 2022-11-25
> > > > > **Thanks for checking our results and considering the score increase**
> > > > >
> > > > > Dear reviewer oMcq,
> > > > >
> > > > > We thank you again for your time to check our response and helping us extend a more general theorem. We appreciate the increase of scores.
> > > > >
> > > > > Best,
> > > > >
> > > > > authors

---

### Official Review · Reviewer_5Wy8 · 2022-11-01

**Confidence:** 2
**Correctness:** 4
**Technical Novelty And Significance:** 3
**Empirical Novelty And Significance:** 3
**Recommendation:** 6

**Clarity, Quality, Novelty And Reproducibility:**

The authors have provided the code to replicate the results. Although the reviewer has not run the code, the implementation looks reasonable at a high level.


**Strength And Weaknesses:**

This is an interesting work with promising results. Would the author please comment on whether the methodology can also be casted as a multi-task learning? Moreover, what are other applications/extensions of the proposed framework? There several discrete black-box expensive potentially noise-corrupted discrete optimization applications which can benefit from this methodology. One such example is chip design and the other is NAS (amongst many other).

**Summary Of The Paper:**

This paper aims to automatically solve combinatorial optimization by leveraging unsupervised learning, learning from historical data and
achieving an instance-wise good solution simultaneously. In order to do so, the authors propose a methodology for warm-starting future combinatorial optimization problem instances. The methodology is based on meta-learning. The authors show that warm-staring is beneficial indeed across multiple datasets and shifts.


**Summary Of The Review:**

The reviewer finds this work interesting and promising with sufficient empirical validation.

---

> ### Author Response · Authors · 2022-11-14
> **Response to reviewer 5Wy8**
>
> Thank you so much for reviewing our paper and your valuable comments. We address and discuss your questions in the following response.
>
> > Would the author please comment on whether the methodology can also be casted as a multi-task learning?
>
> **Reviewer 5Wy8** is so insightful. As there is a neat connection between multi-task learning and meta-learning[1], if we view each instance in CO problems as an independent task, the goal of both meta-learning and multi-task learning is to find good network parameters which work well for multiple tasks. Therefore, our problem formulation can be cast as multi-task learning.
> However, we guess there still exists a slight difference between the two frameworks in implementation. For example, the standard model architecture for multi-task learning requires each task to have a distinct projection head in the final layer following the shared feature extraction layer, which cannot be applied in the standard meta-learning framework.
>
> > There are several discrete black-box expensive potentially noise-corrupted discrete optimization applications which can benefit from this methodology. One such example is chip design and the other is NAS (amongst many others).
>
> This is also a very insightful comment. A promising application of our framework could be black-box discrete optimization such as chip design in circuits or NAS. An interesting further study is to further apply our Meta-EGN framework in these black box CO fields.
>
> [1] Wang et al. Bridging multi-task learning and meta-learning: Towards efficient training and effective adaptation. ICML 2022.

---

### Author Response · Authors · 2022-11-14
**Unified response to all reviewers**

We sincerely thank the reviewers for the time to review our manuscript and their constructive suggestions. Three reviewers (**Reviewer 5Wy8, sUXy, uyeg**) think that our idea is interesting and promising. **Reviewer oMcq** feels confused about some statements of the work and raises insightful questions. We are sorry about the confusion and have tried our best to address it in the revised manuscript.

We would like to point out that although our idea looks simple by combining meta-learning with unsupervised learning for CO, we are the first one to propose this idea and demonstrate its effectiveness in CO problems, which also gets praised by all the reviewers (**Reviewer 5Wy8,oMcq,sUXy,uyeg**). Our idea may further be useful for general objective design of unsupervised learning.

We have introduced the changes in our revised manuscript in the unified response. We will further address other questions in the specific response to each reviewer.
- In Appendix C.3, we separate the algorithm table of Meta-EGN into training and testing procedures, to make it clearer to follow.
- In Appendix C.2, we add the detailed derivation of the three problems and cite the two relative papers inspired by which we relax our loss functions, to make it easier for readers to follow and check.
- In page 7 Table.3,4, we refine the bolded texts to make them satisfy the definition of pareto-optimal solution. The pareto-optimal solution denotes the solutions that: within the time budget, the method achieves the best performance; Or to achieve such performance, the method requires the least time.
- In appendix B.3,4, we add two supplementary experiment results, one is to train the models on only a subset of the full-size training data to check the sensitivity of Meta-EGN, the other is to train the models for the MIS on RRGs with a single degree to check whether the models could still maintain good generalization. The experiment results align with the other results in the paper and the conjecture of **Reviewer sUXy**.
- In appendix B.5 and C.1, we report the wall-clock time for training convergence in different problems on different datasets and add the detailed definition of training epochs for both EGN and Meta-EGN.
- We correct a confusing sentence on page 2.
- We refine some other minor typos as posed by the reviewers.

---

> ### Author Response · Authors · 2022-11-21
> **Unified response to all reviewers 2**
>
> Dear reviewers,
>
> We thank all your efforts and actionable suggestions in reviewing our paper. We would be glad to know if you have any further questions or comments. Thank you very much.
>
> Best,
>
> Authors

---

### Decision · Program_Chairs · 2023-01-20

**Decision:**

Accept: poster

**Justification For Why Not Higher Score:**

The work has some incremental aspects. Some reviewers found the results not particularly surprising (but encouraging).

**Justification For Why Not Lower Score:**

Authors addressed the concerns raised by reviewers in a convincing manner.

**Metareview: Summary, Strengths And Weaknesses:**

This paper presents an interesting application of meta-learning to combinatorial optimization problems. The work is clear and the claims are supported. The conducted benchmarks are convincing. Authors addressed the concerns raised by the reviewers in a satisfactory manner.


**Note From Pc:**

if the above contains the word "oral" or "spotlight" please see: "oral" presentation means -> notable-top-5% and "spotlight" means -> notable-top-25%. As stated in our emails, we are disassociating presentation type from AC recommendations

**Summary Of Ac-Reviewer Meeting:**

N/A